# Numerical modeling of surface wave development under the action of wind

*DMITRY CHALIKOV*

*Shirshov Institute of Oceanology, Saint Petersburg 199053, Russia*

*Russian State Hydrometeorological University, SaintPetersburg195196*

*University of Melbourne, Victoria 3010, Australia*

**Abstract**

The numerical modeling of two-dimensional surface wave development under the action of wind is performed. The model is based on three-dimensional equations of potential motion with free surface written in a surface-following non-orthogonal curvilinear coordinate system where depth is counted from moving surface. Three-dimensional Poisson equation for velocity potential is solved iteratively. Fourier transform method, the second-order accuracy approximation of vertical derivatives on a stretched vertical grid and the fourth-order Runge-Kutta time stepping are used. Both the input energy to waves and dissipation of wave energy are calculated on the basis of the earlier developed and validated algorithms. A one-processor version of the model for PC allows us to simulate an evolution of wave field with thousands of degrees of freedom over thousands of wave periods. A long-time evolution of two-dimensional wave structure is illustrated by the spectra of wave surface and input and output of energy.

## 1. Introduction

Phase resolving modeling of sea waves is the mathematical modeling of surface waves including explicit simulations of surface elevation and velocity field evolution. Compared with the *spectral wave modeling*, the phase resolving modeling is more general since it reproduces a real visible physical process and is based on the well-formulated full equations. The phase resolving models usually operate with a large number of freedom degrees. In general, this method is more complicated and requires more computational resources. The simplest way of such modeling is the calculation of a wave field evolution based on the linear equations. Such approach allows reproducing the main effects of the linear wave transformation due to superposition of wave modes, reflections, refractions etc. This approach is useful for many technical applications while it cannot reproduce a nonlinear nature of waves and the transformation of wave field due to nonlinearity. Another example of a relatively simple object is a case of the shallow-water waves. The nonlinearity can be taken into account in the more sophisticated models derived from the fundamental fluid mechanics equations with some simplifications. The most popular approach is based on nonlinear Schrödinger equation of different orders (see Dysthe, 1979) obtained by expansion of the *surface wave* displacement. This approach is also used for solving the problem of *freak waves*. The main advantage of a simplified approach is that it allows reducing a three-dimensional (3-D) problem to a two-dimensional one (or 2-D problem to 1-D problem). However, it is not always clear which of the non-realistic effects are eliminated or included in the model after simplifications. This is why the most general approach being developed over the past years is based on the initial two-dimensional or three-dimensional equations (still potential). All the tasks based on these

equations can be divided into two groups: the periodic and non-periodic problems. An assumption of periodicity considerably simplifies construction of numerical models though such formulation can be applied to the cases when the condition of periodicity is acceptable, for example, when domain is considered as a small part of a large uniform area. For limited domains with no periodicity the problem becomes more complicated since the Fourier presentation cannot be used directly.

From the point of view of physics, the problem of phase resolving modeling can be divided into two groups: the adiabatic and non-adiabatic modeling. A simple adiabatic model assumes that the process develops with no input or output of energy. Being not completely free of limitations, such formulation allows investigating the wave motion on the basis of true initial equations. Including the effects of input energy and its dissipation is always connected with the assumptions that generally contradict an assumption of potentiality, i.e., new terms added to the equations should be referred to as pure phenomenological. This is why treatment of a non-adiabatic approach is often based on quite different constructions.

All the phase resolving models use the methods of computational mathematics and inherit all their advantages and disadvantages, i.e., on one side, the possibility of a detailed description of the processes, on the other side, a bunch of the specific problems connected with the computational stability, space and time resolution. Mathematical modeling produces tremendous volumes of information the processing of which can be more complicated than the modeling itself.

The phase resolving wave modeling takes a lot of computer time since it normally uses a surface-following coordinate system, which considerably complicates the equations. The most time-consuming part of the model is an elliptic equation for velocity potential usually solved with iterations. Luckily, for a two-dimensional problem this trouble is completely eliminated by use of the conformal coordinates reducing the problem to a one dimensional system of equations which can be solved with high accuracy (Chalikov and Sheinin, 1998). For a three-dimensional problem, the reduction to a two-dimensional form is evidently impossible; hence, the solution of a 3D elliptical equation for velocity potential becomes an essential part of the entire problem. This equation is quite similar to the equation for pressure in a non-potential problem. It follows that the 3-D Euler equations, being more complicated, still can be solved over acceptable computer time.

There is a large volume of papers devoted to the numerical methods developed for investigation of wave processes over the past decades. It includes a Finite Difference Method (Engsig-Karup et al., 2009, 2012), , a Finite Volume Method (Causon et al., 2010), a Finite Element Method (Ma and Yan, 2010; Greaves, 2010), a Boundary (Integral) Element Method (Grue and Fructus, 2010), Spectral Methods (Ducroset et al., 2007, 2012, 2016; Touboul and Kharif, 2010; Bonnefoy et al., 2010). These include a Smoothed Particle Hydrodynamics method (Dalrymple et al., 2010), a Large Eddy Simulation Method (LES) (Issa et al., 2010; Lubin and Caltagirone J.-P. (2010), a Moving Particle Semi-implicit method (Kim et al., 2014), a Constrained Interpolation Profile method (Zhao, 2016), a Method of Fundamental Solutions (Young et al., 2010) and a Meshless Local Petrov–Galerkin method (Ma, 2010). Fully nonlinear model should be applied to many problems.. Most of the models were designed for engineering applications such as overturning waves, broken waves, waves generated by landslides, freak waves, solitary waves, tsunamis, violent sloshing waves, interaction of extreme waves with beaches, interaction of steep waves with the fixed structures or with different floating

structures. The references given above make less than one percent of the publications on those topics

Two-dimensional approach (like conformal method) considers a strongly idealized wave field, since even monochromatic waves in the presence of lateral disturbances quickly obtain a two-dimensional structure. The difficulty arising is not a direct result of increase of dimension. The fundamental complication is that the problem cannot be reduced to a two-dimensional problem, and even for the case of a double-periodic wave field the problem of solution of Laplace-like equation for velocity potential arises. The majority of the models designed for investigation of three-dimensional wave dynamics are based on simplified equations such as the second order perturbation methods in which higher order terms are ignored. Overall, it is unclear which effects are missing in such simplified models.

The most sophisticated method is based on full three dimensional equations and surface integral formulations (Beale, 2001; Xue et al., 2001; Grilli et al., 2001; Clamond and Grue, 2001; Clamond et al, 2005, 2006; Fructus et al., 2005; Guyenne et al., 2006; Fochesato et al., 2006). Fully nonlinear, three-dimensional water waves, which extends an approach was suggested by Craig and Sulem (1993) originally given in the two-dimensional setting. The model is based upon Hamiltonian formulation (Zakharov, 1968) which allows reducing a problem of surface variables computation by introducing Dirichlet–Neumann operator which is expressed in terms of its Taylor series expansion in homogeneous powers of surface elevation. Each term in this Taylor series can be obtained from the recursion formula and efficiently computed using fast Fourier transform.

The main advantage of the boundary integral equation methods (BIEM) is that they are accurate and can describe highly nonlinear waves. A method of solution of Laplace equation is based on use of Green's function, which allows us to reduce a 3-D water-wave problem to a 2-D boundary integral problem. The surface integral method is well suited for simulation of the wave effects connected with very large steepness, specifically, for investigation of a freak wave generation. These methods can be applied both to periodic and non-periodic flows. The methods do not impose any limitations on wave steepness, so they can be used for simulation of the waves that even approach breaking (Grilli et al., 2001) when the surface obtains a non-single value shape. The method allows us to take into account bottom topography (Grue and Fructus, 2010) and investigate an interaction of waves with the fixed structures or with the freely-responding floating structures (Liu et al., 2016, Gou et al., 2010).

However, the BIEM method seems to be quite complicated and time-consuming being applied to a long-term evolution of a multi-mode wave field in large domains. The simulation of relatively simple wave fields illustrates an application of the method, and it is unlikely that the method can be applied to the simulation of a long-term evolution of a large-scale multi-mode wave field with a broad spectrum. Implementation of a multi-pole technique for a general problem of the sea wave simulation (Fochesato et al., 2006) can solve the problem but obviously leads to the considerable algorithmic difficulties.

Currently, the most popular in oceanography approach is a HOS (High Order Scheme) model developed by Dommermuth and Yue (1987); West et al. (1987). The HOS is based on a paper by Zakharov (1968), where a convenient form of the dynamic and kinematic surface conditions was suggested. Equations used by Zakharov were not intended for modeling, but rather for investigation of stability of finite amplitude waves. In fact, a system of coordinates where depth is counted from the surface, was used, but the Laplace equation for velocity

potential was taken in its traditional form. However, the Zakharov's followers have accepted this idea literally. They used two coordinate systems: a curvilinear surface-fitting system for surface conditions and the Cartesian system for calculation of a surface vertical velocity. The analytic solution for velocity potential in the Cartesian coordinate system is known. It is based on the Fourier coefficients on a fixed level, while the true variables are the Fourier coefficients for the potential on a free surface. Here a problem of transition from one coordinate system to another arises. This problem is solved by expansion of the surface potential into the Taylor series in the vicinity of the surface. An accuracy of this method depends on that of representation of an exponential function with a finite number of the Taylor series. For the small-amplitude waves and for a narrow wave spectrum, such accuracy is evidently satisfactory. However, for the case of a broad wave spectrum that contains many wave modes, the order of the Taylor series should be high. The problem is now that the waves with high wave numbers are superposed over the surface of larger waves. Since the amplitudes of a surface potential attenuate exponentially, an amplitude of a small wave at a positive elevation increases, and on the contrary, it can approach zero at negative elevations. It is clear that such setting of HOS model cannot reproduce high-frequency waves, which actually reduces the nonlinearity of the model. This is why such model can be integrated for long periods using no high frequency smoothing. Besides, an accuracy of calculation of vertical velocity on the surface depends on full elevation at each point. Hence, the accuracy is not uniform along a wave profile. A substantial increase of the Taylor expansion order can definitely result in the numerical instability due to occasional amplification of modes with high wave numbers. The authors of a surface integral method shared a similar point of view (Clamond et al., 2005). We should note, however, that comparison of HOS method based on the West et al., (1987) approach using a method of the surface integral for an idealized wave field (Clamond et al., 2006) shows quite acceptable results. It was shown in the previous paper that a method suggested by Dommermuth et al., (1987) demonstrates poorer divergence of the expansion for vertical velocity than for method by West et al., (1987). The HOS model has been widely used (for example, Tanaka, 2001; Toffoli et al., 2010; Touboul and Kharif, 2010) and it has shown its ability to efficiently simulate the wave evolution (propagation, nonlinear wave–wave interactions, etc.) in a large-scale domain (Ducrozet et al., 2007, 2012). It is obvious that the HOS model can be used for many practical purposes. Recently, Ecole Centrale Nantes, LHEEA Laboratory (CNRC) announced that the non-linear wave models based on High-Order Spectral (HOS) are published as an open source (https://github.com/LHEEA/HOS-ocean/wiki).

Opposite to the HOS method based on the analytical solution of Laplace equation in the Cartesian coordinates, a group of models is based on direct solution of the equation for velocity potential in the curvilinear coordinates (Engsig-Karup et al, 2009, 2012; Chalikov et al., 2014). The main advantage of the surface-following coordinate system is that a variable surface is mapped onto the fixed plane. Since wave motion is very conservative, the high accuracy numerical schemes should be used for a good description of nonlinearity and spectrum transformation. This most universal approach is being developed at the Technical University of Denmark (see Engsig-Karup, 2009). Actually, the models *ModelWave3D* developed at TUD are targeted at solution of a variety of problems including such problems as modeling of wave interaction with submerged objects as well as the simulation of wave regime in the basins with real shape and topography.

The model is based on the equations of potential flow with a free surface. An effect of variable bathymetry is taken into account by using a so-called $\sigma$-coordinate,

(straightening out the bottom and surface). At vertical surfaces a normal derivative of the
velocity potential is equal to zero. A flexible-order approximation for spatial derivatives is used.
The most time-consuming part of this mode is a 3-D equation for the velocity potential. The
strategy of the model development is directed at exploiting architectural features of modern
GPUs for the mixed precision computations. This approach is tested using a recently developed
generic library for fast prototyping of PDE (Partial Differential Equations) solvers. The new
wave tool is applicable for solving and analyzing of a variety of large-scale wave problems in the
coastal and offshore engineering. A description of the project and references can be found at site
(http://www2.compute.dtu.dk/~apek/OceanWave3D/).
Comparison of *ModelWave3D* with HOS model was presented by Ducrozet et al., (2012).
It was shown that both model demonstrate high accuracy, but HOS model shows better
performance. Note, that comparison of speed of models in this case is not indicative since
*ModelWave3D* was designed for investigation of complicated processes, taking into account the
real shape of basin, variable depth and even presence of engineering constructions. All these
features obviously are not included in HOS model.
Development of waves under the action of wind is a process that is difficult to simulate
since surface waves are very conservative and change their energy for hundreds and thousands of
periods. This is why the most popular method is spectral modeling. Waves as physical objects in
this approach are actually absent, since evolution of spectral distribution of wave energy is
simulated. The description of input and dissipation in this approach is not connected directly
with the formulation of the problem, but it is rather adopted from other branches of wave theory
where waves are the objects of investigation. However, the spectral approach turned out to be the
only method capable of describing the space and time evolution of wave field in the ocean. The
phase resolving models (or 'direct' models) designed for reproducing waves themselves cannot
compete with spectral models since a typical size of domain in such models does not exceed
several kilometers. Such domain includes just several thousands of large waves. Nevertheless,
direct wave modeling plays an ever-increasing role in geophysical fluid dynamics, because it
gives the possibility to investigate the processes which cannot be reproduced with spectral
models. One of such problems is that of extreme wave generation. (Chalikov, 2009; Chalikov,
and Babanin, 2016a). Direct modeling is also a perfect instrument for development of
parameterization of physical processes for spectral wave models. Besides, such models can be
used for direct simulation of wave regimes of small water basins, for example, port harbors.
Other approaches of direct modeling are discussed in (Chalikov et al., 2014; Chalikov, 2016)
Until recently, direct modeling was used for reproduction of quasi-stationary wave
regime when wave spectrum essentially did not change. An unique example of direct numerical
modeling of surface wave evolution is given in Chalikov and Babanin (2014)where development
of wave field was calculated with use of a two-dimensional model based on full potential
equations written in the conformal coordinates. The model included algorithms for
parameterization of input and dissipation of energy (a description of similar algorithms is given
below). The model successfully reproduced an evolution of wave spectrum under the action of
wind. However, strictly one-dimensional (unidirected) waves are not realistic; hence, the full
problem of wave evolution should be formulated on the basis of three-dimensional equations. An
example of such modeling is given in the current paper.

**2. Equations**

Let us introduce a *non-stationary surface-following non-orthogonal* coordinate system:
$\xi = x, \quad \vartheta = y, \quad \zeta = z - \eta(\xi, \vartheta, \tau), \quad \tau = t$,     (1)
where $\eta(x, y, t) = \eta(\xi, \vartheta, \tau)$ is a moving periodic wave surface given by the Fourier series
$\eta(\xi, \vartheta, \tau) = \sum_{-M_x < k < M_x} \sum_{-M_y < l < M_y} h_{k,l}(\tau) \Theta_{k,l}$,     (2)
where $k$ and $l$ are components of wave number vector k, $h_{k,l}(\tau)$ are Fourier amplitudes for
elevations $\eta(\xi, \vartheta, \tau)$, $M_x$ and $M_y$ are the numbers of modes in directions $\xi$ and $\vartheta$, respectively,
while $\Theta_{k,l}$ are Fourier expansion basis functions, represented as matrix:
$$\Theta_{kl} = \begin{cases} \cos(k\xi + l\vartheta) & -M_x \le k \le M_x, -M_y < l < 0 \\ \cos(k\xi) & -M_x \le k \le 0, l = 0 \\ \sin(k\xi) & 0 \le k \le M_y, l = 0 \\ \sin(k\xi + l\vartheta) & -M_x \le k \le M_x, 0 < l \le M_y \end{cases}$$
(3)

The 3-D equations of potential waves in the system of coordinates (1) at $\zeta \le 0$ take the
following form:
$\eta_\tau = -\eta_\xi \varphi_\xi - \eta_\vartheta \varphi_\vartheta + \left(1 + \eta_\xi^2 + \eta_\vartheta^2\right)\Phi_\varsigma$,     (4)
$\varphi_\tau = -\frac{1}{2}\left(\varphi_\xi^2 + \varphi_\vartheta^2 - \left(1 + \eta_\xi^2 + \eta_\vartheta^2\right)\Phi_\zeta^2\right) - \eta - p$,     (5)
$\Phi_{\xi\xi} + \Phi_{\vartheta\vartheta} + \Phi_{\zeta\zeta} = \Upsilon(\Phi)$,     (6)
where $\Upsilon$ is the operator:
$\Upsilon() = 2\eta_\xi()_{\xi\zeta} + 2\eta_\vartheta()_{\vartheta\zeta} + \left(\eta_{\xi\xi} + \eta_{\vartheta\vartheta}\right)()_\zeta - \left(\eta_\xi^2 + \eta_\vartheta^2\right)()_{\zeta\zeta}$,     (7)
capital fonts $\Phi$ are used for domain $\zeta < 0$ while the lower case $\varphi$ refers to $\zeta = 0$. Term p in (5)
described the pressure on surface $\zeta = 0$.
It is suggested in (Chalikov et al., 2014) that it is convenient to represent velocity
potential $\varphi$ as a sum of two components, i.e., an analytical ('linear') component
$\bar{\Phi}, \left(\bar{\varphi} = \bar{\Phi}(\xi, \vartheta, 0)\right)$ and an arbitrary ('non-linear') component $\tilde{\mathsf{F}}, \left(\tilde{\jmath} = \tilde{\mathsf{F}}(x, \jmath, 0)\right)$:
$\jmath = \bar{\jmath} + \tilde{\jmath}, \quad \mathsf{F} = \bar{\mathsf{F}} + \tilde{\mathsf{F}}$.     (8)
The analytical component $\bar{\Phi}$ satisfies Laplace equation:
$\bar{\Phi}_{\xi\xi} + \bar{\Phi}_{\vartheta\vartheta} + \bar{\Phi}_{\zeta\zeta} = 0$,     (9)
with known solution:
$\bar{\Phi}(\xi, \vartheta, \zeta, \tau) = \sum_{k,l} \bar{\varphi}_{k,l}(\tau) \exp\left(|k|\zeta\right)\Theta_{k,l}$,     (10)
(where $|k| = \left(k^2 + l^2\right)^{1/2}$, $\bar{\varphi}_{k,l}$ are Fourier coefficients of surface analytical potential $\bar{\varphi}$ at $\zeta = 0$).
The solution satisfies boundary conditions:
$\varsigma = 0: \quad \bar{\Phi} = \bar{\varphi}$
$\varsigma \to -\infty: \quad \tilde{\Phi}_\varsigma \to 0$     (11)

The nonlinear component satisfies an equation:
$\tilde{\Phi}_{\xi\xi} + \tilde{\Phi}_{\vartheta\vartheta} + \tilde{\Phi}_{\zeta\zeta} = \Upsilon(\tilde{\Phi}) + \Upsilon(\bar{\Phi})$,     (12)
Eq. (12) is solved with the boundary conditions:

$$\varsigma = 0: \quad \tilde{\Phi} = 0$$
$$\varsigma \to -\infty: \quad \tilde{\Phi}_\varsigma \to 0$$

258                                                                                           (13)

The derivatives of linear component $\bar{\Phi}$ in (7) are calculated analytically. The scheme
combines 2-D Fourier transform method in the 'horizontal surfaces' and a second-order finite-
difference approximation on a stretched staggered grid defined by relation $\Delta\zeta_{j+1} = \chi\Delta\zeta_j$ ($\Delta\zeta$ is
a vertical step, while $j = 1$ at the surface). The stretched grid provides increase of accuracy of
approximation for the exponentially decaying modes. The values of stretching coefficient $\chi$ lie
within the interval 1.01-1.20. A finite-difference second-order approximation of vertical
operators in Eq. (12) on a non-uniform vertical grid is quite straightforward. Equation (12) is
solved as Poisson equations with iterations over the right-hand side. A each time step the
iterations start with right-side calculated at previous time step. The initial elevation was
generated as superposition of linear waves corresponding to JONSWAP spectrum (Hasselmann
et al., 1973[4]) with random phases. The initial Fourier amplitudes for surface potential were
calculated by formulas of linear wave theory. A detailed description of the scheme and its
validation is given in Chalikov et al., (2014) and Chalikov (2016).
Equations (4) – (6) are written in a non-dimensional form by using the following scales:
length $L$ where $2\pi L$ is (dimensional) period in the horizontal direction; time $L^{1/2}g^{-1/2}$ and velocity
potential $L^{3/2}g^{1/2}$ ($g$ is acceleration of gravity). The pressure is normalized by water density, so
that the pressure scale is $Lg$. Equations (4) – (6) are self-similar to the transformation with
respect to $L$. The dimensional size of domain $2\pi L$, so scaled size is $2\pi$. All the results
presented in this paper are nondimensional. Note that the number of Fourier modes can be
different in $x$ and $y$ directions. In this case is assumed that two length scales $L_x$ and $L_y$ are used.
The nondimensional length of domain in $y$-direction remains equal $2\pi$ and factor $r = L_x / L_y$ is
introduced in definition of differentiation in Fourier space.
**3. Energy input and dissipation**
Input energy to waves describes a pressure term $p$ in a dynamic boundary condition (5).
The tangent stress on the surface cannot be taken into account in potential formulation.
Dissipation cannot be also described with use of potential equations, but for realistic description
of wave dynamics, dissipation of wave energy should be taken into account, i.e., we should
include in equations (4) and (5) additional terms which, strictly speaking, contradict the
assumption of potentiality.
**3.1 Energy input from wind**
According to the linear theory(Miles, 1957), the Fourier components of surface pressure
$p$ are connected with those of surface elevation through the following expression:
$$p_{k,l} + \mathrm{i}p_{-k,-l} = \frac{\rho_a}{\rho_w}\left(\beta_{k,l} + \mathrm{i}\beta_{-k,-l}\right)\left(h_{k,l} + \mathrm{i}h_{-k,-l}\right),$$     (14)
where $h_{k,l}, h_{-k,-l}, \beta_{k,l}, \beta_{-k,-l}$, are real and imaginary parts of elevation $\eta$ and the so-called $\beta$-
function (i.e., Fourier coefficients at COS and SIN, respectively); $\rho_a / \rho_w$ is a ratio of air and

water densities. Eq. (14) is a standard presentation of pressure above multi-mode surface. It means that every wave mode with amplitude $\left(h_{k,l}^2 + h_{-k,-l}^2\right)^{1/2}$ initiates the pressure mode with amplitude $\left(p_{k,l}^2 + p_{-k,-l}^2\right)^{1/2}$ shifted by phase of wave mode by angle $\alpha = \operatorname{atan}\dfrac{\beta_{-k,-l}}{\beta_{k,l}}$. Both coefficients in (14) are function of ratio of wind velocity at half of mode length height $\lambda_{k,l}/2$ to virtual phase velocity. Hence, for derivation of shape of beta-function it is necessary to simultaneously measure wave surface elevation and non-static pressure on the surface. Experimental measurement of surface pressure is a very difficult problem since the measurements should be done very close to a moving surface, preferably, with a surface-following sensor. Such measurements are done quite seldom, especially, in the field. The measurements were carried out for the first time by a team of authors both in laboratory and field (Snyder et al, 1981; Hsiao and Shemdin, 1983; Hasselmann and Bösenberg, 1991; Donelan et al., 2005, 2006). The data obtained in this way allowed constructing an imaginary part of beta-function used in some versions of wave forecasting models (Rogers et al., 2012). Such measurements and their processing are quite complicated since wave-produced pressure fluctuations are masked by turbulent pressure fluctuations. The second way of beta-function evaluation is based on the results of numerical investigations of statistical structure of the boundary layer above waves with use of Reynolds equations and an appropriate closure scheme. In general, this method works so well that many problems in the technical fluid mechanics are often solved using numerical models, not experimentally (Gent and Taylor, 1976; Riley et al., 1982; Al-Zanaidi and Hui, 1984).. This method was being developed beginning from (Chalikov, 1978, 1986), followed by (Chalikov and Makin, 1991; Chalikov and Belevich, 1992; Chalikov, 1995). The results were implemented in WAVEWATCH model, i.e., a third-generation wave forecast model (Tolman and Chalikov, 1996) and thoroughly validated against the experimental data in the course of developing WAVEWATCH-III (Tolman et al., 2014). This method was later improved on the basis of more advanced coupled modeling of waves and boundary layer (Chalikov and Rainchk, 2010), while the beta-function used in WAVEWATCH-III was corrected and extended up to high frequencies. Direct calculation of energy input to waves requires both real and imaginary parts of the beta-function. The total energy input to waves depends on imaginary part of $\beta$-function, while the moments of higher order depend both on imaginary and real parts of $\beta$. This is why full approximation constructed in (Chalikov and Rainchik, 2010) was used in the current work. Note that in the range of relatively low frequencies the nw method is very close to the scheme implemented in WAVEWATCH-III.

It is a traditional suggestion that both coefficients are the functions of virtual nondimensional frequency $\Omega = \omega_{k,l} U \cos\psi = U / c_{k,l} \cos\psi$ (where $\omega_{k,l}$ and $U$ are the nondimensional radian frequency and wind speed, respectively; $c_{k,l}$ is a phase speed of the $k^{th}$ mode; $\psi$ is an angle between wind and wave mode directions). Most of the schemes for calculations of $\beta$-function consider a relatively narrow interval of nondimensional frequencies $\Omega$. In the current work, the range of frequencies covers an interval $\left(0 < \Omega < 10\right)$, and occasionally the values of $\Omega > 10$ can appear. This is another reason why the function derived in (Chalikov and Rainchik, 2010) through coupled simulations of waves and boundary layer, is used here. Wave model is based on potential equations for a flow with free surface, extended

with an algorithm for breaking dissipation (see below description of the breaking dissipation
parameterization). Wave boundary layer (WBL) model is based on Reynolds equations closed
with $K-\varepsilon$ scheme; solutions for air and water are matched through the interface. The $\beta$-
function was used for evaluation of accuracy of the surface pressure $p$ calculations. A shape of
$\beta$-function connecting surface elevations and surface pressure, is studied up to high
nondimensional wave frequencies both in positive and negative (i.e., for wind opposite to waves)
domains. The data on $\beta$-function exhibit wide scatter, but since the volume of data was quite
large (47 long-term numerical runs allowed us to generate about 1,400,000 values of $\beta$), the
shape of $\beta$-function was defined with satisfactory accuracy up to very high nondimensional
frequencies $(-50 < \Omega < 50)$. As a result, the data on $\beta$-function in such a broad range, allow us
to calculate wave drag up to very high frequencies and to explicitly divide the fluxes of energy
and momentum transferred by the pressure and molecular viscosity. This method is free of
arbitrary assumptions on the drag coefficient $C_d$; and, on the other hand, such calculations allow
investigating the nature of wave drag (see Ting et al., 2012).
The most reliable data on $\beta$-function are concentrated in interval $-10 < \Omega < 10$ (negative
values of $\Omega$ correspond wave modes running against wind). real and imaginary parts of beta
function are shown in Fig. (1). It is an corrected version of approximation given in Chalikov and
Rainchik (2010), where data at negative $\Omega$ were interpreted erroneously. In current calculations
the modes running against wind are absent.

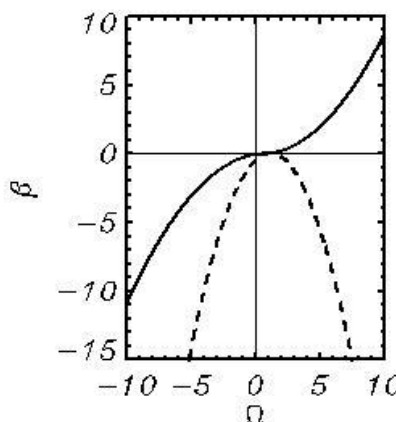

**Figure** 1. Real (dashed curve) and imaginary (solid curve) parts of $\beta$-function.


Function $\beta$ can be approximated by formulas:

$$\beta_{k,l} = \begin{cases} \beta_0 + a_0\left(\Omega - \Omega_0\right) + a_1\left(\Omega - \Omega_0\right)^2 & \Omega_0 < \Omega \\ \beta_0 + a_0\left(\Omega - \Omega_0\right) - a_1\left(\Omega - \Omega_0\right)^2 & \Omega < \Omega_0 \end{cases}, \qquad (15)$$

$$\beta_{-k,-l} = \begin{cases} \beta_1 + a_3\left(\Omega - \Omega_2\right) & \Omega < \Omega_2 \\ a_2\left(\Omega - \Omega_1\right)^2 & \Omega_2 < \Omega < \Omega_3 \\ \beta_1 - a_3\left(\Omega - \Omega_3\right) & \Omega_3 < \Omega \end{cases}, \qquad (16)$$

where the coefficients are:
$\Omega_0 = 0.02277, \Omega_1 = 1.20, \Omega_2 = -18.8, \Omega_3 = 21.2, a_0 = 0.02277, a_1 = 0.09476, a_2 = -0.3718,$
$a_3 = 14.80, b_0 = -0.02, b_1 = -148.0.$

It was indicated above that an initial wave field is assigned as superposition of linear
modes of which amplitudes are calculated with JONSWAP spectrum with initial peak wave
number $k_p^0 = 100$. The initial value $U / c_p^0 = 6$ was chosen, i.e., a ratio of the nondimensional
wind speed at height of half of initial peak wave length $\lambda_0 / 2 = 2\pi / 100$ and the phase speed
$c_p^0 = \left( k_p^0 \right)^{-1/2}$ is equal to 6. Such a high ratio corresponds to initial stages of wave development.
Wind velocity $6c_p^0$ remains constant during all time of integration. The values of $\Omega$ for other
wave numbers are calculated by assuming that wind profile is logarithmic:
$$\Omega_{k,l} = \frac{U}{c_{k,l}} \ln \frac{\lambda_{k,l}}{2z_0} \left( \ln \frac{\lambda_0}{2z_{00}} \right)^{-1} \cos \psi_{k,l} , \tag{17}$$

where $z_{00}$ is effective nondimensional roughness for the initial wind profile, while $z_0$ is the
actual roughness parameter that depends on the energy in a high-frequency part of spectrum and
on the wind profile. We call it 'effective', since very c, lose to the surface the wind profile is not
logarithmic (Chalikov, 1995; Tolman and Chalikov, 1996; Chalikov and Rainchik, 2010). The
value of this parameter depends on the wind velocity and energy in a high-wave number interval
of wave spectrum, as well as on the length scale of the problem. All these effects are possible to
include by matching the wave model with a one dimensional WBL model(Ting et al., 2012).
Here, a simplified scheme for the roughness parameter is chosen. It is well known that the
roughness parameter (as well as a drag coefficient) increases with decrease of the inverse wave
age. In our case wind speed is fixed, and dependence for the nondimensional roughness
parameter is constructed on the basis of the results obtained in (Chalikov and Rainchik, 2010):
$$z_0 = 15z_{00}\Omega , \tag{18}$$

where $z_{00} = 10^{-3}$ is the initial value of the roughness parameter. Eq. (18) approximates
dependence of the effective roughness at the stage of wave development. Note that the results are
not sensitive to variation of the roughness parameter within reasonable limits.

**3.2 High wave number energy dissipation**

A nonlinear flux of energy directed to the small wave numbers produces downshifting of
spectrum, while an opposite flux forms a shape of spectral tail. The second process can produce
accumulation of energy near 'cut' wave number. Both processes become more intensive with
increase of energy input. Growth of amplitudes at high wave numbers is followed by the growth
of local steepness and numerical instability. This well-known phenomenon in numerical fluid
mechanics is eliminated by use of a highly selective filter simulating nonlinear viscosity. To
support stability, additional terms are included into the right hand sides of equations (4) and (5):
$$\frac{\partial \eta_{k,l}}{\partial \tau} = E_{k,l} - \mu_{k,l}\eta_{k,l} , \tag{19}$$

$$\frac{\partial \varphi_{k,1}}{\partial \tau} = F_{k,l} - \mu_{k,l}\varphi_{k,l} \tag{20}$$

( $E_{k,l}$ and $F_{k,l}$ are Fourier amplitudes of the right-hand sides of equations (4) and (5) while factor
$\mu_{k,l}$ is calculated using a formula:
$$\mu_{k,l} = \begin{cases} 0 & |k| < k_d \\ c_m k_0 \left( \dfrac{|k| - k_d}{(k_0 - k_d)} \right)^2 & k_d \leq |k| \leq k_0 \\ c_m k_0 & |k| > k_0 \end{cases} \tag{21}$$

where $k$ and $l$ are components of wave number $|k|$, while coefficients $k_d$ and $k_0$ are defined by the
expressions:
$$k_d = d_m^2 M_x M_y \left( \left( l|k|^{-1} d_m M_x \right)^2 + \left( k|k|^{-1} d_m M_y \right)^2 \right)^{-1/2} \tag{22}$$

$$k_0 = M_x M_y \left( \left( l|k|^{-1} M_x \right)^2 + \left( k|k|^{-1} M_y \right)^2 \right)^{-1/2} \tag{23}$$

where $c_m = 0.1$, $d_m = 0.75$. Expressions (18) - (20) can be interpreted in a straightforward way:
the value of $\mu_{k,l}$ is equal to zero inside the ellipse with semi-axes $d_m M_x$ and $d_m M_y$; then it grows
linearly with $|k|$ up to the value $c_m$ and is equal to $c_m$ outside the outer ellipse. This method of
filtration that we call 'tail dissipation' was developed and validated with a conformal model by
Chalikov and Sheinin (1998). The sensitivity of the results to the parameters in (21) - (23) is not
high. The aim of the algorithm is support of smoothness and monotonicity of wave spectrum
within a high wave number range. Since the algorithm affects the amplitudes of small modes, it
actually does not reduce the total energy, though it efficiently prevents development of
numerical instability. Note that any long-term calculations cannot be performed without 'tail
dissipation' eliminating development of the numerical instability at high wave numbers.

**3.3 Dissipation due to wave breaking**

The main process of wave dissipation is wave breaking. This process is taken into
account in all spectral wave forecasting models similar to WAVEWATCH (see Tolman and
Chalikov, 1996).Since there are no waves in spectral models, no local criteria of wave breaking
can be formulated. This is why breaking dissipation is represented in spectral models in a
distorted form. A real breaking occurs in relatively narrow areas of physical space; however,
spectral image of such breaking is stretched over the entire wave spectrum, while in reality the
breaking decreases height and energy of dominant waves. This contradiction occurs because
waves in spectral models are assumed as linear ones, while in fact the breaking occurs in
physical space with nonlinear sharp wave, usually composed of several modes. However,
progress has been gradually made in spectral wave modeling over the past decade. One important
outcome is that the wave breaking term in the state-of-art wave models now accounts for the threshold-
behavior of dominant wave breaking, that is, waves won't break unless their steepness exceeds a threshold
(Alves and Banner, 2003; Babanin et al., 2010).
The mechanics of wave breaking at developed wave spectrum differs from that in a wave
field represented by few modes, normally considered in many theoretical and laboratory
investigations (e.g., Alberello et al., 2018). Since the breaking in laboratory conditions is initiated
by special assigning of amplitudes and phases, it cannot be similar to the breaking in natural
conditions. To some degree, the wave breaking is similar to development of extreme wave that
appears suddenly with no pronounced prehistory (Chalikov and Babanin, 2016a, 2016b). There
are no signs of modulational instability in both phenomena, which suggests a process of taking
energy from other modes. The evolution leading to breaking or 'freaking' seems just opposite:
full energy of main wave remains nearly constant while the columnar energy is focusing around
the crest of this wave which becomes sharper and unstable. Probably, even more frequent cases
of wave breaking and extreme wave appearance can be explained by local superposition of
several modes.
The instability of interface leading to breaking is an important and poorly developed
problem of fluid mechanics. In general, this essentially nonlinear process should be investigated
for a two-phase flow. Such approach was demonstrated, for example, by Iafrati (2001).
However, the progress in solving this highly complicated problem is not too fast.
The problem of breaking parameterization includes two points: (1) establishing of a
criterion of breaking onset and (2) developing of an algorithm of breaking parameterization. The
problem of breaking is discussed in details in Babanin (2011). Chalikov and Babanin (2012)
performed numerical investigation of the processes leading to breaking. It was found that a clear
predictor of breaking, formulated in dynamical and geometrical terms, probably does not exist.
The most evident criterion of breaking is the breaking itself, i.e., the process when some part of
upper portion of sharp wave crest is falling down. This process is usually followed by separation
of detached volume of liquid into water and air phases. Unfortunately, there is no possibility to
describe this process within the scope of potential theory.
Some investigators suggest using the physical velocity approaching the rate of surface
movement in the same direction as a criterion of breaking onset. This is incorrect, since the
kinematic boundary condition suggests that these quantities are exactly equal to each other. It is
quite clear that the onset of breaking can be characterized by appearance of non-single-value
piece of surface. This stage can be investigated with two-dimensional model which due to a high
flexibility of the conformal coordinates allows us to reproduce a surface with the inclination in
the Cartesian coordinates larger than 90 degrees. (In the conformal coordinates the dependence
of elevation on curvilinear coordinate is always single-value). The duration of this stage is
extremely short, the calculations being always interrupted by the numerical instability with sharp
violation of conservations laws (constant integral invariants, i.e., full energy and volume) and
strong distortion of the local structure of flow. Numerous numerical experiments with conformal
model showed that after appearance of non-single value, the model never returns to stability.
However, introducing of appearance of the non-single-surface as a criterion of breaking
instability even in conformal model is impossible, since a behavior of model at a critical point is
unpredictable, and the run is most likely to be terminated, no matter what kind of
parameterization of breaking is introduced. It means that even in a precise conformal model,
stabilization of solution should be initiated prior to breaking.
Consideration of exact criterion for breaking onset for the models using transformation of
the coordinate type of (1) is useless, since the numerical instability in such models arises not
because of the breaking approaching but because of appearance of large local steepness. Multiple
experiments with direct 3-D wave model show that appearance of local steepness
$\max\left(\partial\eta/\partial x, \partial\eta/\partial y\right)$ exceeding $\approx 2$ (that corresponds to a slope of about 60 degrees) is always
followed by numerical instability but instability can happen far before reaching this value.
Decrease of time step does not make any effect. As seen, a surface with such slope is very far
from being a vertical 'wall', when real breaking starts. However, an algorithm for breaking
parameterization must prevent a numerical instability. The situation is similar to the numerical
modeling of turbulence (LES technique), where the local highly selective viscosity is used to
prevent appearance of too large local gradients of velocity. The description of breaking in direct
wave modeling should satisfy the following conditions. (1) It should prevent the onset of
instability at each point of half million of grid points over more than 100 thousand of time
steps.(2) It should describe in a more or less realistic way the loss of kinetic and potential
energies with preservation of balance between them. (3) It should preserve the volume. It was
suggested in (Chalikov, 2005) that an acceptable scheme can be based on the local highly
selective diffusion operator with special diffusion coefficient. Several schemes of such type were
validated, and finally the following scheme was chosen:
$$\eta_\tau = E_\eta + J^{-1}\left( \frac{\partial}{\partial \xi} B_\xi \frac{\partial \eta}{\partial \xi} + \frac{\partial}{\partial \vartheta} B_\vartheta \frac{\partial \eta}{\partial \vartheta} \right), \tag{24}$$
$$\varphi_\tau = F_\varphi + J^{-1}\left( \frac{\partial}{\partial \xi} B_\xi \frac{\partial \varphi}{\partial \xi} + \frac{\partial}{\partial \vartheta} B_\vartheta \frac{\partial \varphi}{\partial \vartheta} \right), \tag{25}$$
where $F_\eta$ and $F_\varphi$ are the right-hand sides of equations (4) and (5) including the terms introduced
in terms of Fourier coefficients by (19) − (23), $B_\xi$ and $B_\vartheta$ are diffusion coefficients. It was
suggested in the first versions of the scheme that diffusion coefficient depends on a local slope,
however, such scheme did not prove to be very reliable since it did not prevent all of the events
of numerical instability. A scheme based on the calculation of the local curvilinearity $\eta_{\xi\xi}$ and $\eta_{\vartheta\vartheta}$
turned out to be a lot more robust. The calculations of 75 different runs were performed with full
3-D model in (Chalikov et al., 2014) over period of $t = 350$ (70,000 time steps). The total number
of values used for the calculations of dependence in Fig. 2 (thick curve) is about 6 billion. The
normal probability calculated with the same dispersion is shown by thin curve.

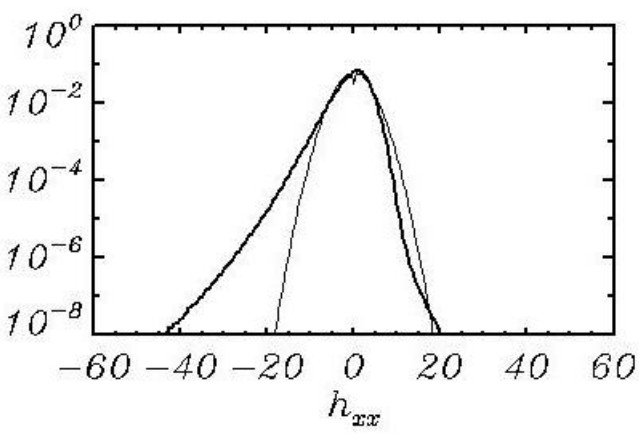

**Figure 2**. Probability of curvilinearity $\eta_{\xi\xi}$ . Thick curve calculated with full 3-D model; thin curve is a probability calculated over ensemble of linear modes with the same spectrum.


It is seen that the probability of large negative values of curvilinearity is by orders larger than the
probability calculated over ensemble of linear modes with spectra generated by nonlinear
model.

The curvilinearity turned out to be very sensitive to the shape of surface. This is why it

was chosen as a criterion of breaking approach. Coefficients $B_\xi$ and $B_\vartheta$ depend nonlinearly on
the curvilinearity
$$B_\xi = \begin{cases} \Delta\xi C_B \eta_{\xi\xi}^2 & \eta_{\xi\xi} < \eta_{\xi\xi}^{cr} \\ 0 & \eta_{\xi\xi} \geq \eta_{\xi\xi}^{cr} \end{cases} \tag{26}$$

$$B_\vartheta = \begin{cases} \Delta\vartheta C_B \eta_{\vartheta\vartheta}^2 & \eta_{\vartheta\vartheta} < \eta_{\xi\xi}^{cr} \\ 0 & \eta_{\vartheta\vartheta} \geq \eta_{\xi\xi}^{cr} \end{cases} \tag{27}$$


where $\Delta\xi$ and $\Delta\zeta$ are horizontal steps in $x$ and $y$ direction in grid space, and coefficients are
$C_B = 2.0$, $\eta_{\xi\xi}^{cr} = \eta_{\vartheta\vartheta}^{cr} = -50$. Algorithm (24) - (27) does not change the volume and decreases the
local potential and kinetic energy. It is assumed that the lost momentum and energy are
transferred to current and turbulence (see Chalikov and Belevich, 1992). Besides, the energy also
goes to other wave modes. The choice of parameters in (24) - (27) is based on simple
considerations: local piece of surface can closely approach the critical curvilinearity but not
exceed it. The values of the coefficients are picked with reserve to provide stability of long runs.
We do not think that the suggested breaking parameterization is a final solution of the
problem. Other schemes will be tried in the next version of the model. However, the results
presented below show that the scheme is reliable and provides a realistic energy dissipation rate.

**4. Calculations and results**

The elevation and surface velocity potential fields are approximated in the current
calculations by $M_x = 256$ and $M_y = 128$ modes in directions $x$ and $y$. The corresponding grid
includes $N_x \times N_y = (1024 \times 512)$ knots. The vertical derivatives are approximated at vertical
stretched grid $d\zeta_{j+1} = \chi d\zeta_j$, $(j = 1, 2, 3 ..., L_w)$ where $\nu = 1.2$ and $L_w = 10$. The small number of
levels used for solution of the equation for nonlinear component of the velocity potential is
possible because just a surface vertical derivative for the velocity potential $\partial\Phi / \partial\zeta (\zeta = 0)$ is
required. The velocity potential mainly consists of an analytical component $\bar\varphi$, while a nonlinear
component provides but small correction. To reach an accuracy of solution $\varepsilon = 10^{-6}$ for equation
(11), no more than two iterations were usually sufficient.
The parameters chosen were used for solution of the problem of wave field evolution
over acceptable time (of the order of 10 days). The initial conditions were assigned on the basis
of empirical spectrum JONSWAP (Hasselmann et al., 1973)with a maximum placed at wave
number $k_p = 100$ with angle spreading $(\cosh\psi)^{256}$. Details of initial conditions are of no
importance because an initial energy level is quite low.
The total energy of wave motion $E = E_p + E_k$ ($E_p$ - is potential energy, while $E_k$ is
kinetic energy) is calculated with the following formulas:

$$E_p = 0.25\overline{\eta^2}, \quad E_k = 0.5\overline{\overline{(\varphi_x^2 + \varphi_y^2 + \varphi_z^2)}}, \tag{28}$$


where single bar denotes averaging over the $\xi$ and $\vartheta$ coordinates, while double bar denotes
averaging over entire volume. The derivatives in (25) are calculated according to transformation
(1). An equation of integral energy $E = E_p + E_k$ evolution can be represented in the following
form:
$$\frac{dE}{dt} = \overline{\overline{I}} + \overline{\overline{D_b}} + \overline{\overline{D_t}} + \overline{\overline{N}},\tag{29}$$

where $\overline{\overline{I}}$ is the integral input of energy from wind (Eqs. (14) – (18); $\overline{\overline{D_b}}$ is a rate of energy
dissipation due to the wave breaking (Eqs. (24) – (27)); $\overline{\overline{D_t}}$ is a rate of energy dissipation due to
filtration of high-wave number modes ('tail dissipation', Eqs. (19) – (23)); $\overline{\overline{N}}$ is an integral effect
of the nonlinear interactions described by the right-hand side of the equations when surface
pressure $p$ is equal to zero. The differential form for calculation of the energy transformation can
be, in principle, derived from Eqs. (4) – (6), but here a more convenient and simple method was
applied. Different rates of integral energy transformations can be calculated with help of
fictitious time steps (i.e., apart from the basic calculations). For example, the value of $\overline{\overline{I}}$ is
calculated by the following relation:

$$\overline{\overline{I}} = \frac{1}{\Delta t}\left(\overline{\overline{E^{t+\Delta t}}} - \overline{\overline{E^t}}\right),\tag{30}$$


where $\overline{\overline{E^{t+\Delta t}}}$ is the integral energy of wave field obtained after one time step with the right side of
equation (6) containing only the surface pressure calculated with Eqs. (14) – (18). For
calculation of the dissipation rate due to filtration, the right-hand side of the equations contains
just the terms introduced in Eqs. (19) - (23), while for calculation of the effects of breaking, only
the terms introduced in (24) – (27) are in use.
An evolution of the characteristics calculated by formula (30) is shown in Fig. 3.

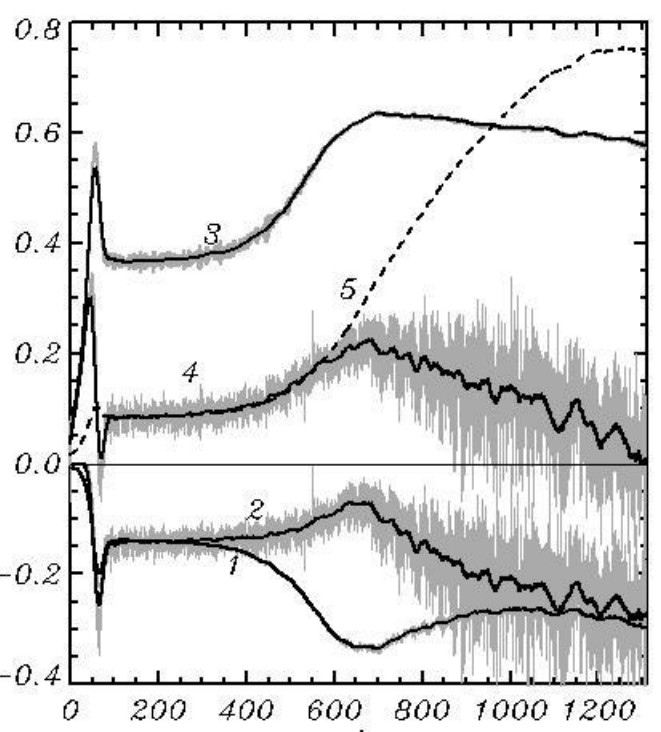

**Figure 3**. Evolution of integral characteristics of solution, rate of evolution of integral energy multiplied by $10^7$) due to: 1 – tail dissipation $D_t$ (Eqs. 19-23); 2 – breaking dissipation $D_b$ (Eqs. 24-27); 3 – input of energy from wind $I$ (Eqs. 14-18); 4 – balance of energy $I + D_t + D_b$. Curve 5 shows the evolution of wave energy $10^5 E$. Vertical bars of grey color show the instantaneous values; thick curve shows the smoothed behavior.

Sharp variations of all characteristics at $t < 50$ can be probably explained by adjustment of linear initial fields to nonlinearity. Up to the end of integration, the sum of all energy transition terms (tail dissipation $\overline{\overline{D}}_t$, breaking dissipation $\overline{\overline{D}}_b$ and energy input $\overline{\overline{I}}$) is approaching zero (curve 4), and the energy growth $E$ (Curve 5) stops. Then the energy tends to decrease, but we are not sure about the nature of this effect. Such behavior can be explained by a fluctuating character of mutual adjustment of input and dissipation or simply by worsening of the approximation because of the downshifting process. Note that opposite to a more or less monotonic behavior of tail dissipation (Curve 1), the breaking dissipation is highly intermittent, which is consistent with common views on the nature of wave breaking.

The data on evolution of wave spectrum are shown in Fig. 4.

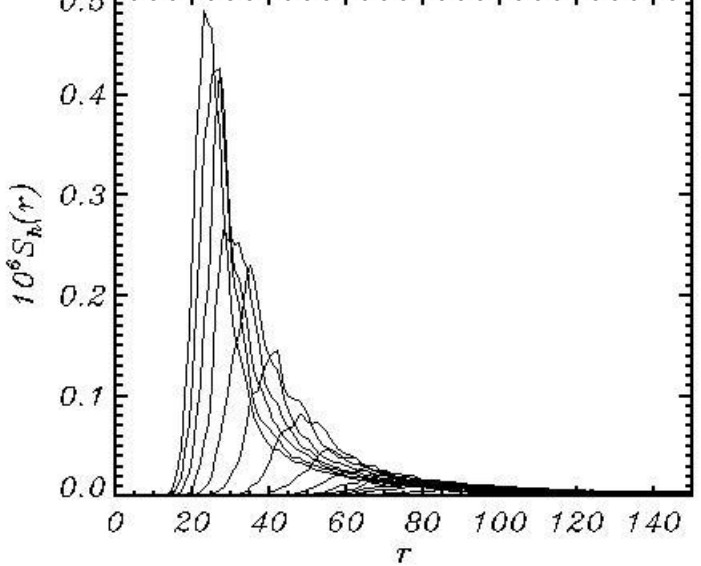

**Figure 4.** The wave spectra $S_h(r)$ integrated over angle $\psi$ in the polar coordinates and averaged over consequent intervals of length about 100 units of nondimensional time $t$. The spectra are growing and shifting from right to left.

The 2-D wave spectrum $S(k,l)$ $\left(0 \le k \le M_x, -M_y \le l \le M_y\right)$ averaged over 13 time intervals of length equal to $\Delta t \approx 100$, was transferred to the polar coordinates $S_p(\psi, r)$ $\left(-\pi/2 \le \psi \le \pi/2, 0 \le r \le M_x\right)$ and then averaged over angle $\psi$ to obtain 1-D spectrum $S_h(r)$:

$$S_h(r) = \sum S_p(\psi, r) r \Delta \psi . \tag{31}$$

An angle $\psi = 0$ coincides with the direction of wind $U$, $\Delta \psi = \pi/180$.

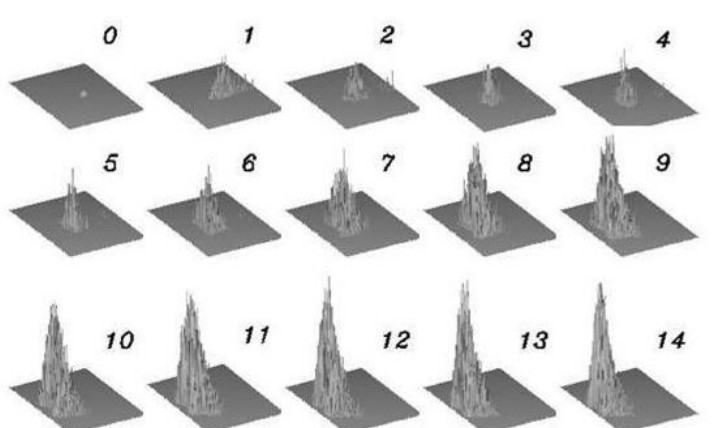

**Figure 5.** Sequence of 3-D images of $\lg_{10}\left(S(k,l)\right)$ where each panel corresponds to single curve in Fig. 3. The left side refers to wave number $l\left(-M_y \le l \le M_y\right)$ and front side – to $k\left(0 \le k \le M\right)$. The numbers indicate

end of time interval expressed in hundreds of nondimensional time units.

As seen, each spectrum consists of separated peaks and holes[1]. This phenomenon was first
observed and discussed by Chalikov et al. (2014). The repeated calculations with different
resolution showed that such structure of 2-D spectrum is typical. It cannot be explained by fixed
combination of interacting modes, since in different runs (with the same initial conditions but
different set of phases for the modes) peaks are located in different locations in Fourier space.

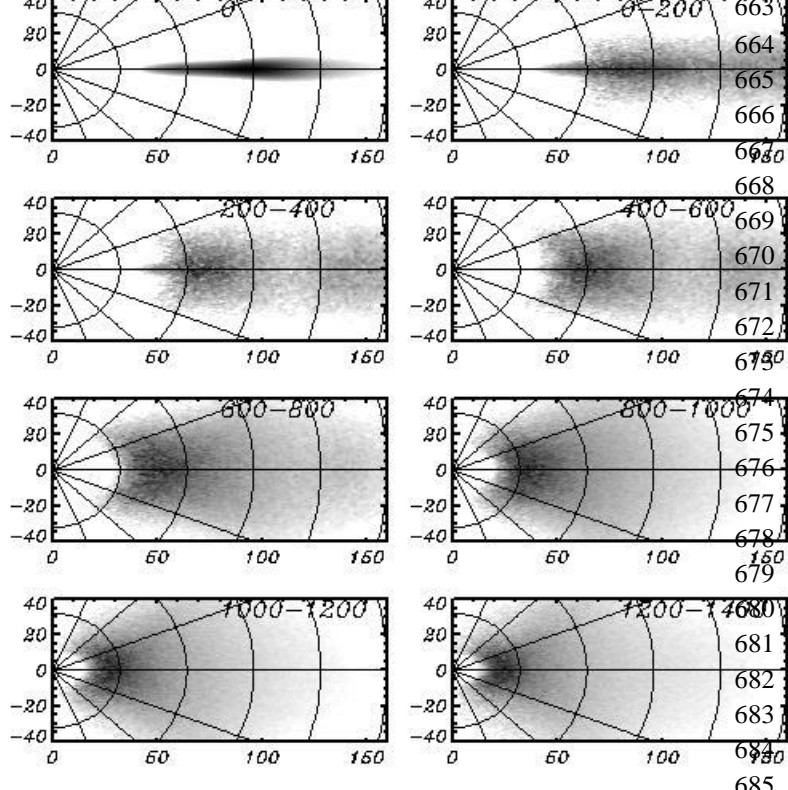

**Figure 6.** Sequence of 2-D images of $\lg_{10}\left(S(k,l)\right)$ averaged over consequent seven periods length $\Delta t = 200$.
Numbers indicate the period of averaging (first panel marked 0, refers to initial conditions). Horizontal and vertical
axes correspond to wave numbers $k$ and $l$ correspondingly

Another presentation is given in Fig 6 where the $\log_{10}\left(S(\psi,r)\right)$, averaged over the successive
seven period length $\Delta t = 200$, is given. The first panel with a mark 0 refers to initial conditions.
Disturbances within the range $(125 < k < 150)$ reflect initial adjustment of the input and
dissipation at high wave number slope of spectrum. The pictures characterize well the
downshifting and angle spreading of spectrum due to nonlinear interactions.
Evolution of the integrated over angles $\psi$ wave spectrum $S_h(r)$ can be described with
the equation

---

[1]The wave spectrum looks rather like La Sagrada Familia (Gaudi) in Barcelona than the St. Mary
Axe ('Pickle' ) in London.

$$\frac{dS_h(r)}{dt} = I(r) + D_t(r) + D_b(r) + N(r), \qquad (32)$$

where $I(r), D_t(r), D_b(r)$ and $N(r)$ are the spectra of the input energy, tail dissipation, breaking
dissipation and a rate of nonlinear interactions, all obtained by integration over angles $\psi$. All of
the spectra shown below were obtained by transformation of 2-D spectra into the polar
coordinate $(\psi, r)$ and then integrated over angles $\psi$ within the interval $(-\pi/2, \pi/2)$. The
spectra can be calculated using an algorithm similar to the algorithm (30) for integral
characteristics. For example, the spectrum of energy input $I(k,l)$ is calculated as follows:
$$I(k,l) = \left( S_c^{t+\Delta t}(k,l) - S_c^t(k,l) \right) / \Delta t, \qquad (33)$$

where $S_c(k_x, k_y)$ is a spectrum of columnar energy calculated by relation
$$S_c(k,l) = \frac{1}{2}\left( h_{k,l}^2 + h_{-k,-l}^2 + \int_{-H}^{0} \left( u_{k,l}^2 + u_{-k,-l}^2 + v_{k,l}^2 + v_{-k,-l}^2 + w_{k,l}^2 + w_{-k,-l}^2 \right) d\zeta \right) \qquad (34)$$

where grid values of velocity components $u, v, w$ are calculated by relations:
$$u = \varphi_\xi + \varphi_\zeta \eta_\xi, \quad v = \varphi_\vartheta + \varphi_\zeta \eta_\vartheta, \quad w = \varphi_\zeta, \qquad (35)$$

and $u_{k,l}, v_{k,l}$ and $w_{k,l}$ are their Fourier coefficients.
For calculation of $I(k,l)$ the fictitious time steps $\Delta t$ are made only with a term
responsible for the energy input, i.e., surface pressure $p$. Spectrum $I(k,l)$ was averaged over the
periods $\Delta t \approx 100$, then transformed into a polar coordinate system and integrated in Fourier
space over angles $\psi$ within the interval $(-\pi/2, \pi/2)$.

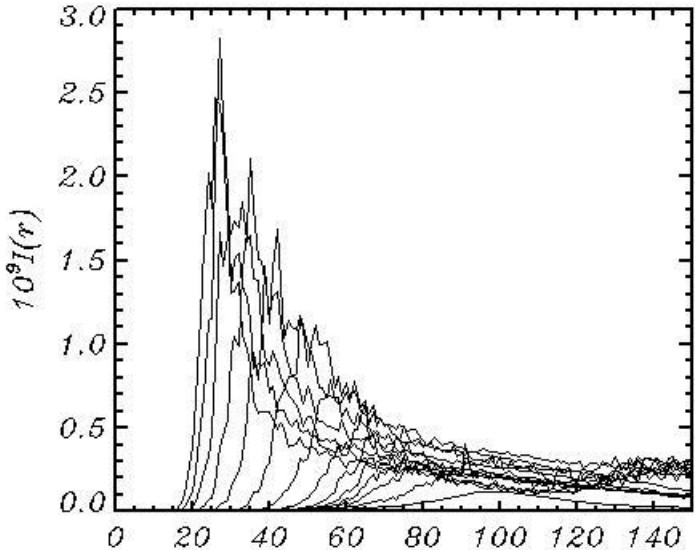

**Figure 7.** The spectrum of energy input $I(r)$ integrated over angle $\psi$ in the polar coordinates and averaged over consequent intervals of length about 100 units of nondimensional time $t$.

The evolution of input spectra (Fig. 7) is, in general, similar to that of wave spectra shown in
Fig. 4. Note that a maximum of spectra is located at the maximum of wave spectra since the
input depends mainly on spectral density, while the dependence on frequency is less important.
Algorithm (32) − (35) was applied for calculation of the dissipation spectra due to
dumping of a high-wave number part of spectrum (tail dissipation) and for calculation of the
spectrum of breaking dissipation. In the first case, the fictitious time step was made taking into
account the terms described by Eqs (19) – (23), while in the second case the time step was made
using the terms described by Eqs (24) – (27).
The spectra of tail dissipation calculated similar to spectra $I(r)$ are shown in Fig. 8.
Dissipation occurs at the periphery of spectrum, outside the ellipse with semi-axes $d_m M_x$ and
$d_m M_y{}^2$. This is why such dissipation, averaged over angles, seems to affect a middle part of 1-D
spectrum. The tail dissipation effectively stabilizes the solution.
The breaking dissipation averaged over angles is presented in Fig. 8. As seen, the
breaking dissipation has a maximum at spectral peak. It does not mean that in the vicinity of
wave peak the probability of large curvilinearity is quite high. The high rate of breaking
dissipation can be explained by high wave energy in the vicinity of wave peak. The energy lost
through breaking, described by the diffusion mechanism, correlates with the energy of breaking
waves. Opposite to high wave number dissipation which regulates shape of spectral tail, the
breaking dissipation forms the main energy-containing part of spectrum.
The diffusion mechanism suggested in (24) - (27) modifies an elevation and surface
stream function in a close vicinity of breaking point. The amplitudes of side perturbation are
small and decrease very quickly over the distance from a breaking point.

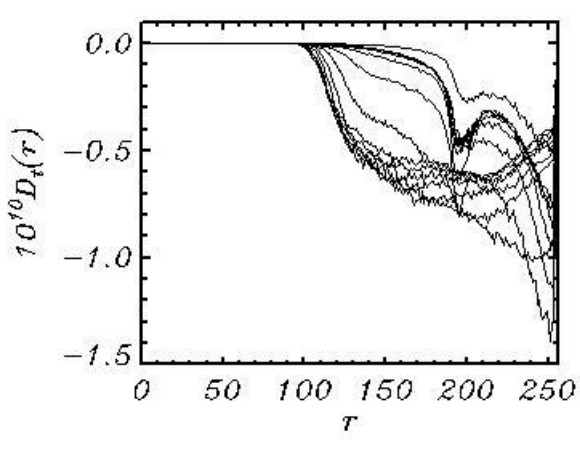

**Figure 8.** Tail dissipation spectra $D_t(r)$ integrated over angle $\psi$ in the polar coordinates and averaged over consequent intervals of length about 100 units of nondimensional time $t$.

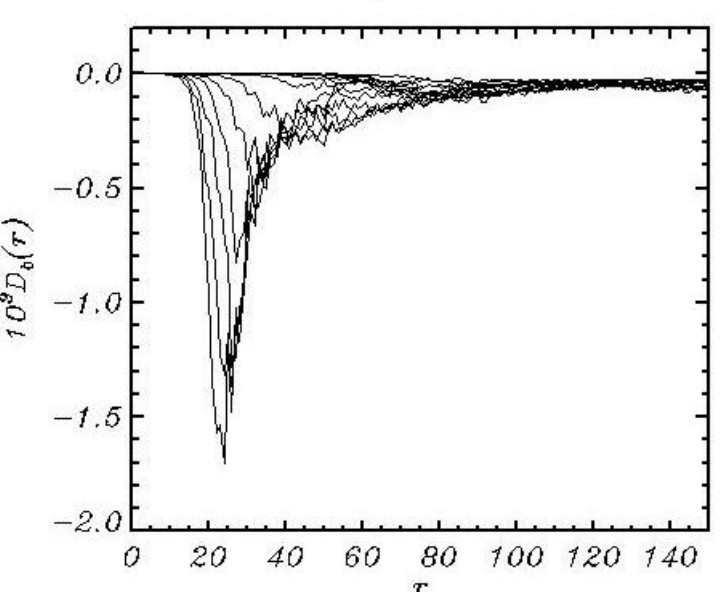

**Figure 9.** Breaking dissipation spectra $D_b(r)$ integrated over angle $\psi$ in the polar coordinates and averaged over consequent intervals of length about 100 units of nondimensional time $t$.

---

[2]The 2-D Fourier spectral 'tail' looks like 'peacock' tail.


An example of profile of the energy input due to breaking $D_b(x)$ is given in Fig. 10. As

seen, energy input is fluctuating around the breaking point. A diffusion operator chosen for
breaking parameterization not only decreases total energy but also redistributes the energy
between Fourier modes in Fourier space.

In general, for the specific conditions considered in the paper, the breaking is an

occasional process taking place in a small part of domain. The kurtosis of input energy due to the
breaking $D_b(\xi, \vartheta)$, i.e., the value
$$Ku = \overline{\overline{D_b^4}} \left( \overline{\overline{D_b^2}} \right)^{-2} - 3 \qquad (36)$$

is of the order of $10^3$, which corresponds to plain function with occasional separated peaks.


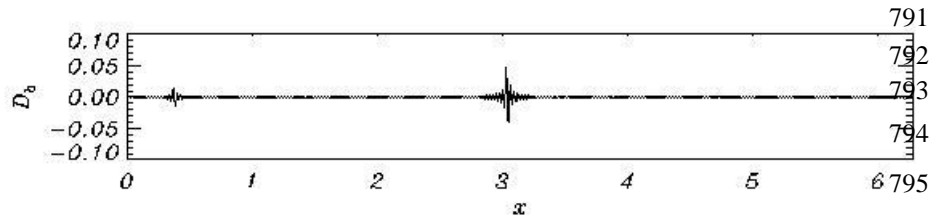

**Figure 10**. Example of
energy input due to
breaking $D_b(x)$.


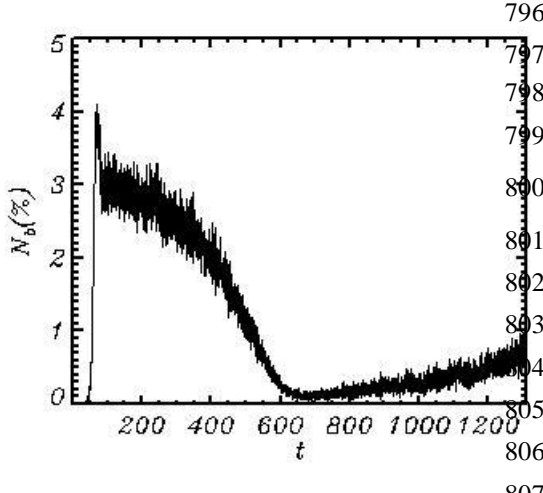


**Figure 11.** Evolution of number of wave breaking
events $N_b$ expressed in percentage of the number
of grid points $N_x \times N_y$.







The number of breaking points in terms of percentage of the total number of points is given in
Fig. 11. As seen, the number of breaking events is going down to $t = 600$ and then is growing up
to the end of the calculations. The number of breaking events is not directly connected with
intensity of breaking, which is seen when comparing Fig. 11 and curve 2 in Fig.3.
An integral term describing nonlinear interaction $\overline{\overline{N}}$ in Eq. (29) is small (compared with local
values of $N_{k,l}$), but the magnitude of spectrum $N(r)$ is comparable with input $I(r)$ and
dissipation $D_t(r)$ and $D_b(r)$ terms. The presentation of term $N(r)$ in a form shown in Figs. (6)
– (8) is not clear. This is why the spectra $10^8 N(r)$ averaged over interval $\Delta t = 100$ are plotted
separately in Fig. 11 for the last eight intervals (thick curves) together with the wave spectrum
$10^6 S_h(r)$. In general, the shapes of spectrum $N(r)$ agree with the conclusions of the quasi-
linear Hasselmann (1962) theory (Hasselmann et al., 1985). At low wave number slope of
spectrum the nonlinear influx of energy is positive while at the opposite slope it is negative. This
process produces shifting of spectrum to the lower wave number (downshifting). Opposite to the
Hasselmann's theory, these results are obtained by solution of full three-dimensional equations.
It would be interesting to compare our results with the calculations of Hasselmann's integral

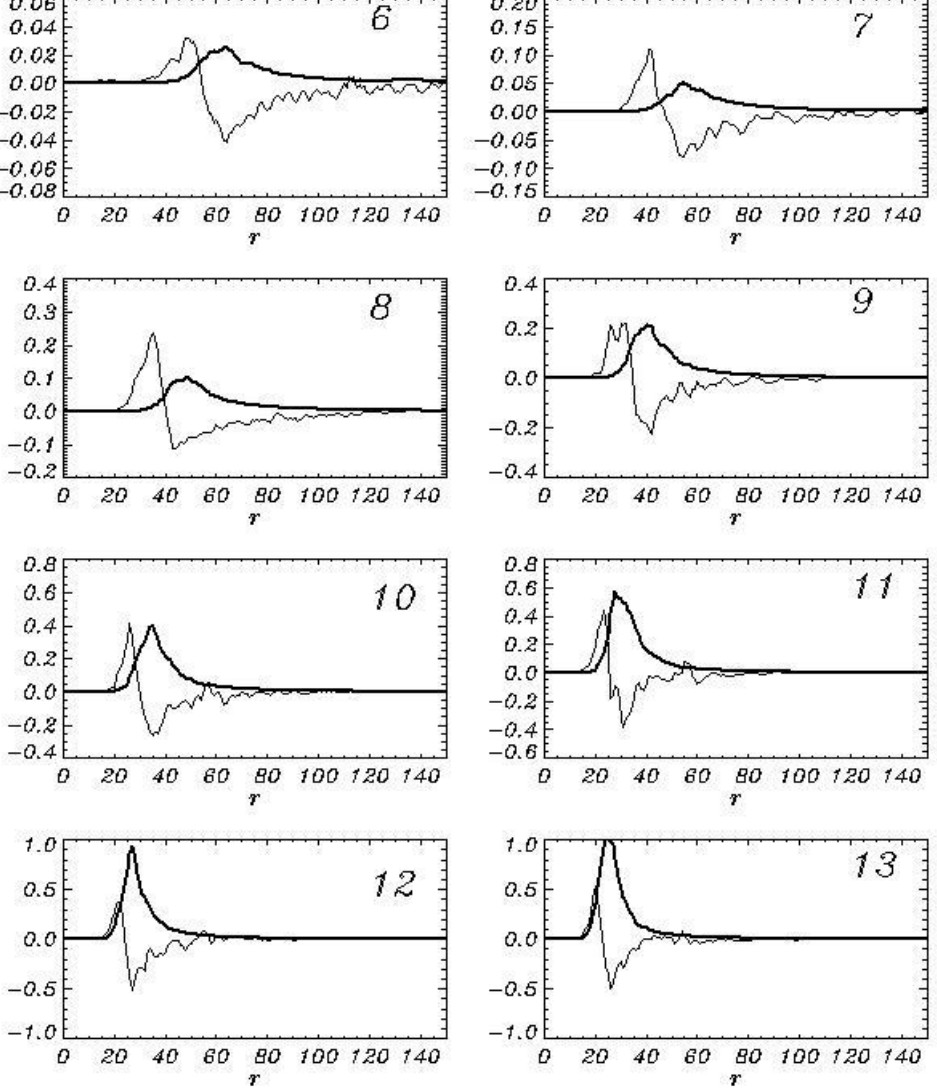

**Figure 12.** Sequence of wave spectra $S_h(r)$ (thick curves) and nonlinear input term $N(r)$ (thin curves) averaged over consequent eight periods of length $\Delta t = 100$ starting from 6[th] period.

Unfortunately, neither of the existing programs of such type permits doing calculations with such
a high resolution that was used in the current model. Note that nonlinear interactions also
produce widening of spectrum.
Obviously, the nonlinearity is quite an important property of surface waves. The
contribution of nonlinearity can be estimated, for example, by comparison of the kinetic energy
of linear component $E_l = 0.5\overline{\overline{\left(\overline{\varphi}_x^2 + \overline{\varphi}_y^2 + \overline{\varphi}_z^2\right)}}$ and the total kinetic energy $E_k$ (Fig. 13). A ratio
$E_l / E_k$ as a function of time remains very close to 1, which proves that the nonlinear part of
energy makes up just a small percentage of the total energy. It does not mean that the role of
nonlinearity is small; its influence can manifest itself over large time scales.



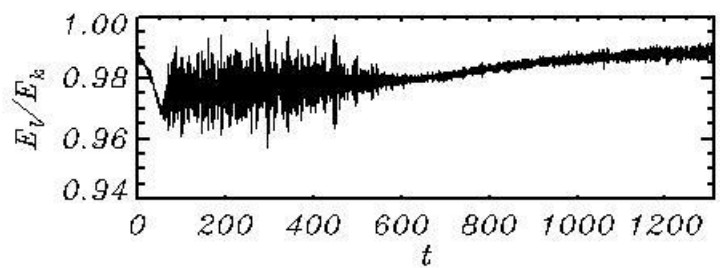

**Figure 13.** Time evolution of ratio $E_l / E_k$ .

The time evolution of integral spectral characteristics is presented in Fig. 14.

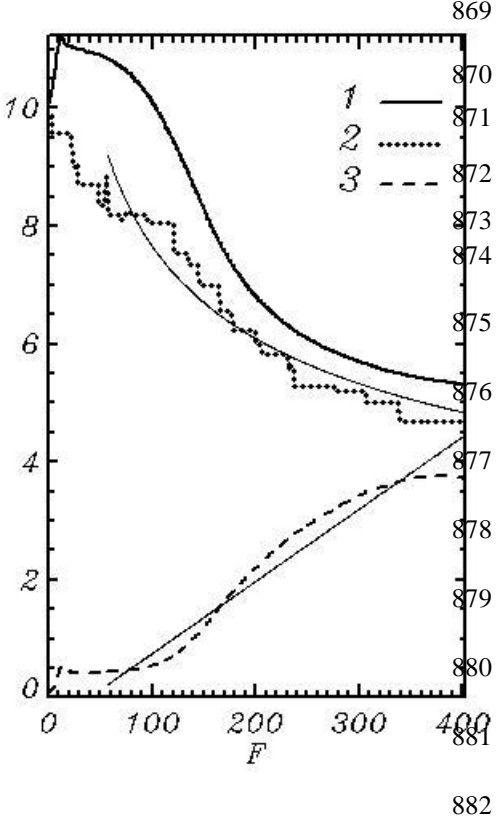

**Figure 14.** Time evolution of: weighted frequency
$\omega_w$      (1) (Eq. 34); spectral peak frequency
$\omega_p = k_p^{1/2}$ (2); full energy $E$    (3) (Eq. 28). Thin
curves are empirical a distance passed by the spectral
peak.


Curve 1 corresponds to the weighted frequency $\omega_w$

$$\omega_w = \left( \frac{\int kSdkdl}{\int Sdkdl} \right)^{1/2} , \tag{37}$$

where integrals are taken over the entire Fourier-domain. The value $\omega_w$ is not sensitive to the
details of spectrum, hence, it well characterizes the position of spectrum and its shifting. Curve2
describes evolution of the spectral maximum. The step shape of curve corresponds to the
fundamental property of downshifting. Opposite to the common views, development of spectrum
occurs not monotonically, but by appearance of a new maximum at lower wave number as well
as by attenuation of the previous maximum. . Interesting that the same phenomenon is also
observed in spectral model (Rogers et al., 2012). Curve 3 describes the change of total energy
$E = E_p + E_k$ . As seen all three curves have a tendency for slowing down  of evolution rate. Then
the energy tends to decrease, but we are not sure about the nature of this effect. Such behavior
can be explained by a fluctuating character of mutual adjustment of input and dissipation or

simply by worsening of the approximation because of the downshifting process. The numerical experiment reproduces the case when development of wave field occurs under the action of permanent and uniform wind. This case corresponds to JONSWAP experiment. Despite large scatter, the data allow us to construct empirical approximations of wave spectrum, as well as to investigate the evolution of spectrum as a function of fetch $F$. In particular, it is suggested that the frequency of spectral peak changes as $F^{-1/3}$, while full energy grows linearly with $F$. Neither of the dependences can be exact, since they do not take into account the approaching to a stationary regime. Besides, the dependence of frequency on fetch is singular at $F = 0$.

The value of fetch in periodic problem can be calculated by integration of peak phase velocity $c_p = |k|^{-1/2}$ over time.

$$F = \int_{t_0}^{t} c_p \, dt \tag{38}$$

The JONSWAP dependencies for the frequency of spectral peak $\omega_p$ and full energy $E$ are shown in Fig 14 by thin curves. Dependence $\omega_p \sim F^{1/3}$ is qualitatively valid. Dependence of the total energy on fetch does not look like a linear one, but it is worth to note that JONSWAP dependence is evidently inapplicable to a very small and large fetch.

## 5. Discussion

A model based on the full three-dimensional equations of potential motion with free surface was used for simulation of development of wave fields. The model is written in the surface-following nonstationary non-orthogonal coordinate system. The details of numerical scheme and the results of validation of the model were described in (Chalikov et al., 2014). The main difference between the given model and HOS model (Ducroset et al., 2017) is that our model is based on direct solution of 3-D equations for velocity potential. This approach is similar to that developed at Technical University of Denmark (TUD, see Engsig-Karup et al., 2009). Actually, the models developed at TUD are directed to solution of a variety of problems including such problems as modeling of wave interaction with submerged objects and simulation of wave regime in the basins with real shape and topography.

In the current paper a three-dimensional model was used for simulation of development of wave field under the action of wind and dissipation. The input energy is described by single term, i.e., surface pressure $p$ in Eq. (4). It is traditionally assumed that the complex pressure amplitude in Fourier space is linearly connected with the complex elevation amplitude with a complex coefficient called $\beta$ – function. Such simple formulations can be imperfect. Firstly, it is assumed that wave field is represented by superposition of linear modes with slowly changing amplitudes and phase velocity obeying the linear dispersive relation. This assumption is valid only for a low-frequency part of spectrum. In reality, the amplitudes of medium and high-frequency modes undergo fluctuations created by reversible interactions. A solid dispersion relation does not connect their phase velocities with wave number. Besides, it is also quite possible that the suggestion of linearity of the connection between pressure and elevation amplitudes is not precise, i.e., $\beta$ – function can depend on amplitudes of modes.

We are not familiar with any observational data that can be used for formulation of
statistically provided scheme for calculation of the input energy to waves. The only method that
can give more or less reliable results is mathematical modeling of the statistical structure of
turbulent boundary layer above a curvilinear moving surface, of which characteristics satisfy
kinematic conditions. The method described above is based on several millions values of
pressure referred strictly to surface. As a whole, the problem of boundary layer seems even more
complicated than the wave problem itself. Some early attempts to solve this problem were made
on the basis of the finite difference two-dimensional model of boundary layer written in the
simple surface following coordinate (see review Chalikov, 1986). Waves were assigned as a
superposition of linear modes with random phases corresponding to the empirical wave
spectrum. This approach was not quite accurate since it did not take into consideration the
nonlinear properties of surface (for example, the sharpness of real waves and the absence of
dispersive relation for waves of medium and high frequencies. The next step was formulation of
coupled models for boundary layer and potential waves, both written in the conformal
coordinates (Chalikov and Rainchik, 2014). The calculations showed that pressure field consists
mostly of random fluctuations not directly connected with waves. A small part of these
fluctuations is in phase with surface disturbances. The calculated values of $\beta$ in Eq. (13) have
large dispersion. However, since the volume of data was very large, the shape of $\beta$-function was
found with high-level accuracy. Probably, approximation of $\beta$ used in the current work can be
considered as most adequate. We are planning additional investigations based on coupled wind-
wave models.  The next step in investigations of Wave boundary Layer (WBL)should use a
three-dimensional LES approach. Note that even availability of large volume of data on the
structure of WBL does not make the problem of parameterization of wind input in spectral wave
models easily solvable, since the pressure is characterized by a broad continuous spectrum
created by nonlinearity.
The wave breaking is obviously even more complicated than the input energy.
Nevertheless, this problem can be simplified, if common ideas used in the numerical fluid
mechanics are accepted. For example, in LES modeling the more or less artificial viscosity is
introduced to prevent too large local velocity gradients. It is a fact that the numerical instability
terminating computations precedes wave breaking. Hence, the scheme should prevent breaking
approach to preserve stability of the numerical scheme. Hence, a wave model should contain the
algorithms preventing appearance of too large slopes. The criterion of breaking is introduced not
for recognizing of the breaking itself, but for the choice of places where it might happen (or,
unfortunately, might not happen). Finally, the algorithm should produce local smoothing of
elevation (and surface potential). The algorithm should be highly selective so that 'breaking'
would occur within narrow intervals and not affect the entire area. The exact criteria of breaking
events (most evident of them is the breaking itself) cannot be used for parameterization of
breaking since in coordinate system (1) the numerical instability occurs long before breaking. In
our opinion, the most sensitive parameter indicating potential instability is the curvilinearity
(second derivative) of elevation.
In the current work, the breaking is parameterized by diffusion algorithm with the
nonlinear coefficient diffusion providing high selectivity of smoothing. We admit that such
approach can be realized in many different forms. The same situation is observed in a problem of
turbulence modeling for parameterization of subgrid scales. Note, that breaking dissipation in
phase resolving models is included in more realistic manner, that in spectral models. For

example, the breaking is simulated in physical space, what allows to reduce the height and energy of nonlinear waves, composed of several modes, In spectral models dissipation is distribute more or less arbitrarily over entire spectrum. Spectral models sometimes include the additional dissipation of short waves due to their modulation by long waves (Young and Babanin, 2006; Babanin et al., 2010). In phase resolving models this process have been included explicitly.

We can finally conclude that the physics included in the wave model is still based on a shaky ground. Nevertheless, the result of the calculations looks quite realistic, which convinces us that the approach deserves further development.

The numerical models of waves similar to that considered in the paper have a lot of important applications. Firstly, they are a perfect tool for development of physical parameterizations schemes in spectral wave models. Secondly, the direct model can be used in future for numerical simulation of wave processes in the basins of small and medium size. These investigations can be based on HOS model (Ducrozet et al., 2016) or the model used in the current paper. However, the most universal approach seems to be developed at the Technical University of Denmark (see Engsig-Karup, 2009). Any model used for a long-term simulation of wave field evolution should include the algorithms describing transformation of energy, similar to those considered in the current paper.

### Acknowledgements

The authors thank Mrs. O. Chalikova for her assistance in preparation of the manuscript as well as anonymous reviewers for their constructive comments. The investigation is supported by Russian Science Foundation, Project 16-17-00124.

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

**Figure captions**

**Figure** 1. Real (dashed curve) and imaginary (solid curve) parts of $\beta$-function.

**Figure 2**. Probability of curvilinearity $\eta_{\xi\xi}$ . Thick curve calculated with full 3-D model; thin curve is a probability calculated over ensemble of linear modes with the same spectrum.

**Figure 3**. Evolution of integral characteristics of solution, rate of evolution of integral energy multiplied by $10^7$) due to: 1 – tail dissipation $D_t$ (Eqs. 19-23); 2 – breaking dissipation $D_b$ (Eqs. 24-27); 3 – input of energy from wind $I$ (Eqs. 14-18); 4 – balance of energy $I + D_t + D_b$. Curve 5 shows the evolution of wave energy $10^5 E$. Vertical bars of grey color show the instantaneous values; thick curve shows the smoothed behavior.

**Figure 4.** The wave spectra $S_h(r)$ integrated over angle $\psi$ in the polar coordinates and averaged over consequent intervals of length about 100 units of nondimensional time $t$. The spectra are growing and shifting from right to left.

**Figure 5.** Sequence of 3-D images of $\lg_{10}(S(k,l))$ where each panel corresponds to single curve in Fig. 3. The left side refers to wave number $l(-M_y \le l \le M_y)$ and front side – to $k(0 \le k \le M)$. The numbers indicate end of time interval expressed in hundreds of nondimensional time units.

**Figure 6.** Sequence of 2-D images of $\lg_{10}(S(k,l))$ averaged over consequent seven periods length $\Delta t = 200$. Numbers indicate the period of averaging (first panel marked 0, refers to initial conditions). Horizontal and vertical axes correspond to wave numbers $k$ and $l$ correspondingly

**Figure 7.** The spectrum of energy input $I(r)$ integrated over angle $\psi$ in the polar coordinates and averaged over consequent intervals of length about 100 units of nondimensional time $t$.

**Figure 8.** Tail dissipation spectra $D_t(r)$ integrated over angle $\psi$ in the polar coordinates and averaged over consequent intervals of length about 100 units of nondimensional time $t$.

**Figure 9.** Breaking dissipation spectra $D_b(r)$ integrated over angle $\psi$ in the polar coordinates and averaged over consequent intervals of length about 100 units of nondimensional time $t$.

**Figure 10**. Example of energy input due to breaking $D_b(x)$.

**Figure 11.** Evolution of number of wave breaking events $N_b$ expressed in percentage of the number of grid points $N_x \times N_y$.

**Figure 12.** Sequence of wave spectra $S_h(r)$ (thick curves) and nonlinear input term $N(r)$ (thin curves) averaged over consequent eight periods of length $\Delta t = 100$ starting from $6^{\text{th}}$ period.

**Figure 13.** Time evolution of ratio $E_l / E_k$

**Figure 14.** Time evolution of: weighted frequency $\omega_w$ (1) (Eq. 34); spectral peak frequency $\omega_p = k_p^{1/2}$ (2); full energy $E$ (3) (Eq. 28). Thin curves are empirical a distance passed by the spectral peak.