# Peer review of "Numerical modeling of surface wave development under the action of wind"

_Ocean Science, 2018_

## Referee Comment (RC1) · Anonymous Referee #1 · 23 Mar 2018

**Review on the manuscript**
**"Numerical modeling of surface wave development under the action of wind"**
**by D. Chalikov submitted to Ocean Science Journal**

In the paper the author suggests a self-consistent phase-resolving numerical model based on the potential Euler equations, aiming to reproduce the processes of generation and evolution of sea wind waves. In some sense such models should replace the ones based on the kinetic theory if the computer performance allows it ever in future. The model is still much simplified, though in principle could be further improved. Most important, the author presents his conceptual ideas on the potential possibilities of the approach and its hopeless sides. The weak points are honestly summarized in the concluding section. The author is a very qualified researcher in this topic, and I was reading the paper with much interest. Therefore I do suggest publishing this paper in the journal.

However there are some critical remarks and some of them are of the general kind. The review of existing literature on the direct simulations of wind waves and corresponding models is obviously distorted as only the papers published by the author are cited. The works by Ducrozet *et al* and by Engsig-Karup *et al* are mentioned in the concluding remarks only. The author of the manuscript is definitively aware of other works (e.g., 3D simulations were performed by G. Ducrozet, A. Toffoli, C. Viotti, W. Xiao, etc., with coworkers). The introduction must give a relatively broad view on the ongoing research in the world scientific community, but in the present form it does not.

I am always wondering why the figures prepared by this author are of such horrible quality. It is still possible to read them, though they look marginally acceptable to the present-day standards. In a few pieces the text looks rather slipshod and not sufficiently proofread.

**Comments**

Sec. 2: The letter $k$ is used for different terms throughout the text (index in Eq. (2), modulus of the wave vector in Eq. (9), component of the wave vector in Eq. (19)), what is confusing (especially in Sec. 3.2). The misprint in line 516 probably suggests a better nomenclature for the wave vector components, $\mathbf{k} = (k_x, k_y)$; the indices may be obviously renamed.

It is not absolutely clear, are $\Theta_{k,j}$ just constants or functions of time.

The velocity potentials are functions of time, what is not shown.

The variable $p$ is not designated for the pressure clearly.

line 89: Eq. (10) contains boundary conditions, thus the phrase "second-order approximation of Eq. (10)" is not clear.

lines 93-97. The general discussion of self-similarity of the solution is fine, though the use of scale $L$ is not clear at this point. It is important to say that periodic boundary conditions are implied, and hence the natural scale $L$ appears. The size of the scaled domain is then $2\pi$.

Eq. (13) and the corresponding text. I suggest rewriting this block in more detail. It is difficult to understand what all these notations mean. I assume, the pressure should be proportional to the gradient of the surface displacement, what is not seen from (13). Besides, the function $\beta$ is discussed but not presented (formula or figure required).

line 135: the abbreviation CR for a reference is introduced, but is not used regularly in what follows. I believe this abbreviation is not necessary.

line 143: the low index at $\Omega$ may be probably omitted.

line 161: "It was indicated above..." – it was not, as far as I can see.

line 164: what is $\lambda_0$?

line 176: "decreases with decrease of the inverse wave age" should probably be changed to "decreases with increase of the wave age".

line 228-234: the similarity between breaking waves and freak waves is unclear and far from the topic. I suggest removing this discussion.

Eqs. (21)-(22) are written in a different style compared to Eqs. (16)-(17) (see the low indices); should be rewritten in a similar format.

line 313: should be "$\Delta\xi$ and $\Delta\vartheta$" instead of "$\Delta\xi$ and $\Delta\zeta$".

line 352: please correct "to33filtration".

Fig. 2: a transitional stage is clearly seen at $t \approx 50$. What is the reason for it?

Fig. 4, caption: the last line should probably read "side – to $k$ $(0 \le k \le M)$" rather than "side – to $k$ $(-M \le k \le M)$".

Fig. 3, 4, 11: they would better have the labels which indicate the dimensional time instants (or time intervals) rather than the number in the sequence.

Fig. 5: please give the values of $k_0$ and $k_d$ for these simulations.

lines 551-552: If I understand the discussion correctly, the dissipation at the middle part in Fig. 7 should be absent if $d_m M_x < d_m M_y$, right?

line 632-633: the sentence sounds absurd, or the issue is not clear. On the one hand, $\overline{\overline{N}}$ is small in physical space (compared to what?), on the other hand its spectral counterpart $N(r)$ is not. Please reformulate or clarify.

Fig. 12: I do not have clear idea why this figure and the corresponding discussion are necessary.

Eq. (34): as far as I understand, it is the definition of $\omega_w$, not $\omega_p$.

line 709: "$k_w$" should be replaced with "$\omega_w$", I assume.

Fig. 13 caption and the corresponding discussion: it is not clear if $\omega_p$ is calculated directly from the frequency spectrum, or according to the dispersion relation, $\omega_p = k_p^{1/2}$?

line 715: "As seen all three curves have a tendency for saturation". According to the note in lines 394-396, this saturation is not necessarily due to the approaching to equilibrium. It would be helpful to repeat this comment here.

lines 738, 805: The references to Ducrozet et al, 2016 have typos in names and years.

---

## Referee Comment (RC2) · Anonymous Referee #2 · 9 Apr 2018

Review of the article titled "*Numerical modeling of surface wave development under the action of wind*" by Dr. Dmitry Chalikov (2018)

**General Comments:**

Recommendation: **Minor Revision**

The manuscript presents simulations of two-dimensional wave fields under steady and homogeneous winds. The wind input term $S_{in}$ or $\beta$ was obtained from a wave-boundary layer model which was established by using Reynolds equations in Chalikov and Rainchik (2011). The kinematics and dynamics at the sea side, i.e., below the air-sea interface, was simulated with three-dimensional equations of potential motion. Within this direct wave model, dissipation of wave energy consists of two terms: i) wave breaking as represented by a local highly selective diffusion operator, of which the diffusion coefficients depend on the local curvilinearity; ii) another selective filter which is only applicable to high wavenumbers. The redistribution of wave energy over spectral space as a result of nonlinear interactions is directly described by the potential equations. Numerical experiments suggest that this wave model is able to yield realistic evolution of wave spectrum. First, the downshifting of the spectral peak and the angular spreading as wave develops are well simulated. Second, the shape of the nonlinear term, modelled by this direct model, is in a good agreement with Hasselmann's integral. Third, the simulated peak frequency $\omega_p$ and wave energy $E$, as a function of fetch, are comparable to measurements collected in the JONSWAP project. Discussion about the disadvantage/weakness of this wave model is also presented.

The direct wave model described in the manuscript is very unique, and the results shown here are quite encouraging. The reviewer is very impressed by Fig. 11 that nonlinear terms from the author's model is qualitatively similar to solutions from Hasselmann's equation.

There are, however, also some details that need to be clarified by the author:

- To suppress the numerical instability at spectral tail, the author employed a filtration term which only applies to high wavenumbers. In my opinion, an alternative method to dissipate energy at high wavenumbers might be to include the induced breaking of short waves by long waves. The field study by Young and Babanin (2006) found the dominant wave breaking can induce the dissipation of wave components at high frequencies. The latest spectral wave models use a wave breaking term $S_{ds}$ consisting of two components: a) inherent breaking when waves are too steep, b) induced breaking due to the modulation of long waves. It's interesting to see how this induced breaking mechanism can be applied in the author's direct model in the future. At this stage, the author may just discuss about this briefly in the manuscript if possible. See, for example,

Donelan, M., 2001: A nonlinear dissipation function due to wave breaking. Ocean Wave Forecasting, 2-4 July 2001, ECMWF, Shinfield Park, Reading, ECMWF, 87-94.

Young, I. R., and A. V. Babanin, 2006: Spectral Distribution of Energy Dissipation of Wind-Generated Waves due to Dominant Wave Breaking. Journal of Physical Oceanography, 36 (3), 376-394.

Babanin, A. V., K. N. Tsagareli, I. R. Young, and D. J. Walker, 2010: Numerical Investigation of Spectral Evolution of Wind Waves. Part II: Dissipation Term and Evolution Tests. Journal of Physical Oceanography, 40 (4), 667-683.

Donelan, M.A., Curcic, M., Chen, S.S., Magnusson, A.K., 2012. Modeling waves and wind stress. J. Geophys. Res. 117 (November 2011), C00J23. doi:10.1029/2011JC007787.

- The English and the quality of figures need to be improved. All the figures in the manuscript don't look very clear. The author may enhance the resolution (dpi) of those figures, or alternatively use vector format such as pdf, eps. Besides, the format of a number of citations in the text is not correct. I have pointed out some, but not all, of them.

**Specific Comments:**

L18: "thousands degrees" to "thousands of degrees"

L25-26: "hundreds and thousands periods" to "hundreds and thousands of periods"

L31: "capable to describe" to "capable of describing"

L32: "that of extreme wave generation" to "the generation of extreme waves"?
L32: "Chalikov, Babanin, 2016a" to "Chalikov and Babanin, 2016a"

L38: "a perfect instrument" to "an useful instrument". I think here the word "perfect" is too strong.

L41: "in (Chalikov et al. 2014, Chalikov, 2016)" to "in Chalikov et al. (2014) and Chalikov (2016)"

L43: "A unique example" to "An unique example"

L44: "in (Chalikov and Babanin, 2014)" to "in Chalikov and Babanin (2014)"

L59: the meaning of $h_{k,l}(\tau)$ as wave mode amplitude is not mentioned here.

L68: Please mention explicitly that $\Phi$ or $\varphi$ is velocity potential.

L112: delete "respectively"

L120: I think it is necessary to cite the WAM work here as it is the first third-generation wave model and also uses a parameterization of $S_{in}$ based on a field experiment conducted by Synder et al. (1981).

The WAMDI Group, 1988. The WAM model - a third generation ocean wave prediction model. J. Phys. Oceanogr. 18 (12), 1775-1810.

L123: "so well" to "reasonably well"

L124-126: It also might be useful to mention works by some other researchers, such as Gent and Taylor (1976), Riley et al (1982), Al-Zanaidi and Hui (1984).

L144: "This is why the function derived in (Chalikov and Rainchik, 2010)" to "This is another reason why the function derived in CR"

L159: "on the contrary" to "on the other hand"

L162: "modes which amplitudes" to "modes of which amplitudes"

L171: "Tolman, Chalikov" to "Tolman and Chalikov, 1996"

L173: "to include" to "to be included"

L161-182: The symbol $\Omega_0$ in Eq. (14) and $\Omega$ in Eq. (15) are quite confusing. It's not clear to me that which wave mode is using when $\Omega_0$ or $\Omega$ is calculated. When the author said $\Omega_0 = 6$, it appears that the initial peak wavenumber of the JONSWAP spectrum is used, i.e., $k_p = 100$. Besides, when the author said "In our case wind speed is fixed", it is unclear that at which height wind speed is fixed. Is it $U_{10}$, that is the wind speeds at 10 m above the sea surface? Please clarify these details if possible.

L190: "This phenomenon well known" to "This well-known phenomenon"

L218-219: "Since there are no waves in spectral models, no local criteria of wave breaking can be formulated."

Just a short comment here: Progress has been gradually made in spectral wave modelling over the past decade. One important outcome is that the wave breaking term $S_{ds}$ in the state-of-art wave models now accounts for the threshold-behavior of dominant wave breaking, that is, waves won't break unless their steepness exceeds a threshold. The saturation spectrum $B(f) = k^3 F(k)$ is used to quantify the *local* steepness of each wave component. See, for example,

Alves, J. H. G. M., and M. L. Banner, 2003: Performance of a Saturation-Based Dissipation-Rate Source Term in Modeling the Fetch-Limited Evolution of Wind Waves. Journal of Physical Oceanography, 33, 1274-1298.

Babanin, A. V., K. N. Tsagareli, I. R. Young, and D. J. Walker, 2010: Numerical Investigation of Spectral Evolution of Wind Waves. Part II: Dissipation Term and Evolution Tests. Journal of Physical Oceanography, 40 (4), 667-683.

L227: "... many theoretical and laboratory investigations (e.g., Alberello et al., 2018)"

Alberello, A., A. Chabchoub, J. P. Monty, F. Nelli, J. H. Lee, J. Elsnab, and A. Toffoli, 2018: An experimental comparison of velocities underneath focussed breaking waves. Ocean Engineering, 155, 201-210.

L287-288: It might be necessary to explain explicitly that $B_\xi$ and $B_\vartheta$ are diffusion coefficients. And for L288: "the first versions" to "the first version"

L329: "$d\zeta_{j+1} = vd\zeta_j$" — It may be better to use the symbol $\chi$ instead of $v$ here for the consistency with L86 where $\Delta\zeta_{j+1} = \chi\Delta\zeta_j$ is used.

L352: "'to33filtration' to "to filtration"

L367-402: The author mentioned in L632 that $\overline{\overline{N}}$ in RHS of Eq. (26) is very small. Is it possible to show the evolution of $\overline{\overline{N}}$ in Fig. 2? I expect $\overline{\overline{N}}$ is almost zero and does not vary with time as the nonlinear interaction only redistributes energy over spectral space and does not change the wave energy of the entire volume.

L429: "calculated by averaging ... 100 units of nondimensional time $t$" — This was already mentioned in L424-426. So maybe just simply say "The resulting wave spectra $S_h(r)$ are presented in Fig. 3."

L550-560: From Fig. 7 and Fig. 8 we know that the tail dissipation $D_t(r)$ is comparable to or even higher than the breaking dissipation $D_b(r)$. Is this an expected behavior of this wave model?

L611: Please clarify what is $x$ in Fig. 9.

L632-675: It's very impressive that the shape of $N(r)$ shown in Fig. 11 is in good agreement with Hasselmann's integral. It also might be useful to mention that Hasselmann's integral exhibits another positive lobe at high frequencies. See, for example,

Hasselmann, S., K. Hasselmann, J. H. Allender, and T. P. Barnett, 1985: Computations and Parameterizations of the Nonlinear Energy Transfer in a Gravity-Wave Spectrum. Part II: Parameterizations of the Nonlinear Energy Transfer for Application in Wave Models. Journal of Physical Oceanography, 15 (11), 1378-1392.

L707-709: the use of the symbols $\omega_p$ and $k_w$ are not correct here as the author intends to say $\omega_w$.

L711-714: the step shape of the curve for $\omega_p$ could be possibly resulted from the discrete nature of wave models and the method utilized to calculate $\omega_p$. Rogers et al. (2012, JTech) showed the similar step-shaped evolution of peak period $T_p$ (see their Fig. 3).

Rogers, E. W., A. V. Babanin, and D. W. Wang, 2012: ObservationConsistent Input and Whitecapping Dissipation in a Model for WindGenerated Surface Waves: Description and Simple Calculations. Journal of Atmospheric and Oceanic Technology, 29 (9), 1329-1346.

L727: "the wavenumber of spectral peak $k_p$" — Please use peak frequency $\omega_p$ for consistency with the caption of Fig. 13.

L734: "three-dimensional equations potential motion" to "three-dimensional equations of potential motion"

L756: "any observation data" to "any observational data"

L759: "which characteristics" to "of which characteristics"

L756-759: This argument appears too strong as $\beta$ measured from field experiments, such as the one proposed by Donelan et al. (2006), is also proved well-performed in operational forecasts/hindcasts.

L764-767: This sentence does not read well. It sounds like "This approach was quite accurate " due to some drawback/weakness. Please reword it, and the right bracket ")" is missing in the end of this sentence.

L768: "(Chalikov, Rainchik, 2014)" to "(Chalikov and Rainchik, 2014)"

L805: "(Ducroset et al. 216)" to "(Ducroset et al. 2016)". Besides, the year shown here and in the References list is not consistent with L738 "(Ducroset et al. 2017)".

Please correct them if necessary.

---

## Author Comment (AC2) · 19 Apr 2018

**Reply to review 2.**

**To suppress the numerical instability at spectral tail, the author employed a _ltra-
tion term which only applies to high wavenumbers. In my opinion, an alternative
method to dissipate energy at high wavenumbers might be to include the induced
breaking of short waves by long waves. The _eld study by Young and Babanin
(2006) found the dominant wave breaking can induce the dissipation of wave com-
ponents at high frequencies. The latest spectral wave models use a wave breaking
term S$_{ds}$ consisting of two components: a) inherent breaking when waves are too
steep, b) induced breaking due to the modulation of long waves. It's interesting
to see how this induced breaking mechanism can be applied in the author's direct
model in the future. At this stage, the author may just discuss about this briefly
in the manuscript if possible.**

It is interesting suggestion. I am not satisfied with current scheme of high-frequency dissipation.
However, I believe that mechanism descibed is already included in models, since the model is phase
resolving one and interaction of long and short waves is simulated directly. Long waves produce the
modulations of short waves and approach them to breaking which is simulated by current algorithm. In
spectral model such interaction between long and short waves is absent, so this dissipation should be
included by hands.
Anyway, it worth to think about this process and methods of special processing for evaluating of
importance f this sort of dissipation.

The amendment made in Lns. 992-998

All technical comments have been taken into account.

---

## Author Response (AR1)

**Reply to review 1.**

I am grateful Reviewer 1 for many value comments and apologize for multiple misprints and discrepancies. In a new version of paper all comments will be taken into account. The numbers of lines are given for improved version reviewer's comments (in Review also) are given by bold fonts.

**However there are some critical remarks and some of them are of the general kind. The review of existing literature on the direct simulations of wind waves and corresponding models is obviously distorted as only the papers published by the author are cited. The works by Ducrozet *et al* and by Engsig-Karup *et al* are mentioned in the concluding remarks only. The author of the manuscript is definitively aware of other works (e.g., 3D simulations were performed by G. Ducrozet, A. Toffoli, C. Viotti, W. Xiao, etc., with coworkers). The introduction must give a relatively broad view on the ongoing research in the world scientific community, but in the present form it does not.**

I extended significantly the review devoted to 3-D modeling. There are so many works (several hundred on my estimate), that it is practically impossible even to mention all of them. I include the papers whose authors made codes by themselves (like me) but not just used them.
The Lns. 25-195 (3 pages) with review have been added as well as 32 new references to papers devoted to 3-D direct modeling.

**I am always wondering why the figures prepared by this author are of such horrible quality. It is still possible to read them, though they look marginally acceptable to the present day standards. In a few pieces the text looks rather slipshod and not sufficiently proofread**

The pictures lose quality at adjusting of size in version prepared for reviewers and at making pdf online which produces errors. My own pdf is excellent as well as original pictures. I use IDL8.6.1 what gives high quality -better than MATLAB. Anyway, sorry for that.

**Sec. 2: The letter *k* is used for different terms throughout the text (index in Eq. (2), modulus of the wave vector in Eq. (9), component of the wave vector in Eq. (19)), what is confusing (especially in Sec. 3.2). The misprint in line 516 probably suggests a better nomenclature for the wave vector components, k = (*kx*, *ky*); the indices may be obviously renamed.**

There was mismatch with wave numbers. All designations have been checked and corrected:
*k, l* are components of wave number vector $\mathbf{k}$, the module of $\mathbf{k}$ is $|k| = \left( k^2 + l^2 \right)^{1/2}$

**It is not absolutely clear, are $\Theta_{k,l}$ just constants or functions of time.**

Definition of function $\Theta_{k,l}$ is inserted (Eq. 3)

**The velocity potentials are functions of time, what is not shown.**

Time $\tau$ is inserted in Eq. (2).

**The variable $p$ is not designated for the pressure clearly.**

The comment is given in Lns 242,243

**line (?): Eq. (10) contains boundary conditions, thus the phrase "second-order approximation of Eq. (10)" is not clear.**

**lines 93-97. The general discussion of self-similarity of the solution is fine, though the use of scale $L$ is not clear at this point. It is important to say that periodic boundary conditions are implied, and hence the natural scale $L$ appears. The size of the scaled domain is then $2\pi$.**

The explanations are added in Lns 279,280

**Eq. (13) and the corresponding text. I suggest rewriting this block in more detail. It is difficult to understand what all these notations mean. I assume, the pressure should be proportional to the gradient of the surface displacement, what is not seen from (13). Besides, the function $\beta$ is discussed but not presented (formula or figure required).**

The Reviewer discusses the next piece of text

*According to the linear theory(Miles, 1957), the Fourier components of surface pressure $p$ are connected with those of surface elevation through the following expression:*

$$p_{k,l} + \mathrm{i}p_{-k,-l} = \frac{\rho_a}{\rho_w}\left(\beta_{k,l} + \mathrm{i}\beta_{-k,-l}\right)\left(h_{k,l} + \mathrm{i}h_{-k,-l}\right), \qquad\qquad 14?)$$

*where $h_{k,l}, h_{-k,-l}, \beta_{k,l}, \beta_{-k,-l}$, are real and imaginary parts of elevation $\eta$ and the so-called $\beta$-function (i.e., Fourier coefficients at COS and SIN, respectively); $\rho_a / \rho_w$ is a ratio of air and water densities, respectively. Hence, for derivation of shape of beta-function it is necessary to simultaneously measure wave surface elevation and non-static pressure on the surface.*

The pressure is NOT proportional to gradient of surface; the structure of pressure above waves is much more complicated.

Eq. (14) is a standard presentation of pressure above multi-mode surface. It means that every wave mode with amplitude $\left(h_{k,l}^2 + h_{-k,-l}^2\right)^{1/2}$ (i.e., coefficients at COS and SIN) initiates the

pressure mode with amplitude $\left( p_{k,l}^2 + p_{-k,-l}^2 \right)^{1/2}$ shifted by phase of wave mode by angle

$\alpha = \operatorname{atan} \dfrac{\beta_{-k,-l}}{\beta_{k,l}}$. Both coefficients in (14) are function of ratio of wind velocity at half of mode

length to virtual phase velocity .

Suggestion to calculate surface pressure as proportional to local inclination is not supported by any experimental data and theory. For example the flow above steep wave generates long positive disturbance of pressure in wave trough and narrow minimum of negative pressure just above the wave peak.

The interaction of wave field with turbulent wind is a subject of special branch of geophysical fluid mechanics which is from point of view of numerical modeling is more complicated than wave modeling.

The additional explanations of algorithm for energy input inserted in text in Lns (?=?). The approximation of $\beta$ -function and Figure have been added.

Note, that correct description of pressure term is crucially important for formation of wave spectrum and long-term wave dynamics.

**line 135: the abbreviation CR for a reference is introduced, but is not used regularly in what follows. I believe this abbreviation is not necessary.**

Accepted.

**line 143: the low index at · may be probably omitted**

$\Omega$ is supplied with indexes when spectral values are considered. $\Omega$ with no indexes is just name of argument in function $\beta$ .

**line 161: "It was indicated above..." – it was not, as far as I can see.**

Inserted in Lns **(267-270)**:

The initial elevation was generated as superposition of linear waves corresponding to JONSWAP spectrum (Hasselmann et al, 19734) with random phases. The initial Fourier amplitudes for surface potential were calculated by formulas of linear wave theory.

**line 164: what is $\lambda_0$ ?**

Text is corrected in Lns 377-383

...height of half of peak wave length...

**line 176: "decreases with decrease of the inverse wave age" should probably be changed to "decreases with increase of the wave age".**

Correctedin Ln. 391

**line 441-447: the similarity between breaking waves and freak waves is unclear and far from the topic. I suggest remove and references ng this discussion.**

I prefer to leave this sentence (together with references) stating that modulational instability is not so general as many people believe. It was shown in several my papers.

**Eqs. (24)-(25) are written in a different style compared to Eqs. (4)-(5) (see the low indices); should be rewritten in a similar format.**

These are the same equation in differential form as Eqs (4) and (5) but simplified by abbreviations. To indicate difference with Eqs. (4), (5) inserted in Lns. 506, 507

**...** introduced in terms of Fourier coefficients by $(16) - (20)$.

**line 527: should be $\Delta\xi$ and $\Delta\vartheta$ instead of $\Delta\xi$ and $\Delta\zeta$.**

Corrected

**line 352: please correct "to33filtration".**

Corrected

**Fig. 3: a transitional stage is clearly seen at $t \sim 50$. What is the reason for it?**

I do not know exactly. However the sentence has been included in Lns 613, 614

 Sharp variations of all characteristics at $t < 50$ can be probably explained by adjustment of linear initial  fields to nonlinearity.

**Fig. 5, caption: the last line should probably read "side $k\left(0 \le k \le M\right)$" rather than "side $-k\left(-M \le k \le M\right)$"**

Sure. Corrected.

**Fig. 4, 5, 12: they would better have the labels which indicate the dimensional time instants (or time intervals) rather than the number in the sequence.**

*There is too small space for printing time intervals. The comment inserted in Fig. caption:*

'The numbers indicate end of time interval expressed in hundreds of nondimensional time units.'

*Since the time is anyway conventional the hundreds of unit are not worse than single unit.*

**Fig. 6: please give the values of $k_0$ and $k_d$ for these simulations.**

I cannot give the values of $k_0$ and $k_d$ since they are functions of $k$ and $l$.

**lines 743-745: If I understand the discussion correctly, the dissipation at the middle part in Fig. 7 should be absent if $d_m M_x < d_m M_y$, right?**

No, the high-wave-number dissipation exists everywhere outside of ellipse which is smaller than Fourier domain.

**line 825,826: the sentence sounds absurd, or the issue is not clear. On the one hand, $N$ is small in physical space (compared to what?), on the other hand its spectral counterpart $N(r)$ is not. Please reformulate or clarify.**

There was no words 'physical space'

An integral term describing nonlinear interaction $\overline{\overline{N}}$ in Eq. (26) is small, but the magnitude of spectrum $N(r)$ is comparable with input $I(r)$ and dissipation $D_t(r)$ and $D_b(r)$ terms. Nonlinear interactions produces exchange by energy between modes.

$\overline{\overline{N}}$ is the integral of the rate nonlinear interaction while, $N(r)$ is rate of nonlinear interactions averaged over angles.

Theoretically, $\overline{\overline{N}}$ is equal 0 but it is correct in infinitely large Fourier domain. Since our Fourier domain is restricted, $\overline{\overline{N}}$ is not zero, and dominantly negative (it could be sometimes positive because of errors of scheme). Anyway, $\overline{\overline{N}}$ is small. However, $N(r)$ is function of $r$ and it is not small, while integral of $N(r)$ over $r$ should be equal to $\overline{\overline{N}}$.

Inserted in Ln. ?:

'(compared with local values in Fourier space of $N_{k,l}$)'

**Fig. 12: I do not have clear idea why this figure and the corresponding discussion are necessary.**

On my opinion it is very interesting picture. People talk a lot on nonlinearity, but nobody actually knows, what is the energy of nonlinear component. For example, if somebody doing the phase resolving simulation of wave regime in harbor, it would be useful to estimate the ratio of $\omega$ nonlinear and linear component. If this value is very small, it is not reasonable to spend the electric energy for exact simulation: linear equations gives all effects of superposition, reflection, refraction, etc...
Small nonlinear part indicates that nonlinearity manifests itself only on large time and space scales.

**Eq. (37): as far as I understand, it is the definition of $\omega_w$ , not $\omega_p$ .**

Sure. Corrected.

**Fig. 14 caption and the corresponding discussion: it is not clear if $\omega_p$ is calculated directly from the frequency spectrum, or according to the dispersion relation.**

$\omega_p = k_p^{1/2}$ is inserted in Figure caption.

Of course the frequency should be product of solution, but for low wave number the dispersion relation is valid with high accuracy. This is not for tail: large time scatter is observed for $\omega$.

**line 7910: "As seen all three curves have a tendency for saturation". According to the note in lines 611,614, this saturation is not necessarily due to the approaching to equilibrium. It would be helpful to repeat this comment here.**

Agree. Comment inserted.

**Reply to Review 2.**

**To suppress the numerical instability at spectral tail, the author employed a tradition term which only applies to high wavenumbers. In my opinion, an alternative method to dissipate energy at high wavenumbers might be to include the induced breaking of short waves by long waves. The _eld study by Young and Babanin (2006) found the dominant wave breaking can induce the dissipation of wave components at high frequencies. The latest spectral wave models use a wave breaking term Sds consisting of two components: a) inherent breaking when waves are too steep, b) induced breaking due to the modulation of long waves. It's interesting to see how this induced breaking mechanism can be applied in the author's direct model in the future. At this stage, the author may just discuss about this briefly in the manuscript if possible. See, for example,**
**1**
**Donelan, M., 2001: A nonlinear dissipation function due to wave breaking. Ocean Wave Forecasting, 2-4 July 2001, ECMWF, Shin_eld Park, Reading, ECMWF, 87-94.**
**Young, I. R., and A. V. Babanin, 2006: Spectral Distribution of Energy Dissipation of Wind-Generated Waves due to Dominant Wave Breaking. Journal of Physical Oceanography, 36 (3), 376-394.**
**Babanin, A. V., K. N. Tsagareli, I. R. Young, and D. J. Walker, 2010: Numerical Investigation of Spectral Evolution of Wind Waves. Part II: Dissipation Term and Evolution Tests. Journal of Physical Oceanography, 40 (4), 667-683.**
**Donelan, M.A., Curcic, M., Chen, S.S., Magnusson, A.K., 2012. Modeling waves and wind stress. J. Geophys. Res. 117 (November 2011), C00J23. doi:10.1029/2011JC007787.**

It is interesting suggestion. I am not satisfied with current scheme of high-frequency dissipation. However, I believe that mechanism described above is already included in models, since the model is phase resolving one and interaction of long and short waves is simulated directly. Long waves produce the modulations of short waves and approach them to breaking which is simulated by current algorithm. In spectral model such interaction between long and short waves is absent, so this dissipation should be included by hands.

Anyway, it worth to think about this process and methods of special processing for evaluating of importance of this sort of dissipation for including the algorithm in spectral models.

The amendment included in Lns. 983-986

**The English and the quality of figures need to be improved. All the figures in the manuscript don't look very clear. The author may enhance the resolution (dpi) of those figures, or alternatively use vector format such as pdf, eps.**

The pictures lose quality at adjusting of size in version prepared for reviewers and at making pdf online which produces errors. My own pdf is excellent as well as original pictures. I use IDL8.6.1 what gives high quality - better than MATLAB. Anyway, sorry for that.

**L38: "a perfect instrument" to "an useful instrument". I think here the word 'perfect" is too strong.**

The problem is that geophysical fluid mechanics is falling behind technical FM. Currently, laboratory equipment in many places just dismantling because modeling gives better quality and much cheaper data. As one of the authors of approach I prefer term 'perfect'.

**L59: the meaning of hk;l(_ ) as wave mode amplitude is not mentioned here.**

Inserted, Lns. 231-233

**L120: I think it is necessary to cite the WAM work here as it is the first third-generation wave model and also uses a parameterization of Sin based on a field experiment conducted by Synder et al. (1981).**
**The WAMDI Group, 1988. The WAM model - a third generation ocean wave prediction model. J. Phys. Oceanogr. 18 (12), 1775-1810.**

During my work at KNMI I knew that scheme for input energy in WAMDI was formulated not on basis of Snyder's data but on semi-analytic and very primitive model by Janssen, what I am not going to mention.

**L124-126: It also might be useful to mention works by some other researchers, such as Gent and Taylor (1976), Riley et al (1982), Al-Zanaidi and Hui (1984).**

Done. Lns. 314,315

**L161-182: The symbol 0 in Eq. (14) and in Eq. (15) are quite confusing. It's not clear to me that which wave mode is using when 0 or is calculated. When the author said 0 = 6, it appears that the initial peak wavenumber of the JONSWAP spectrum is used, i.e., kp = 100. Besides, when the author said \In our case wind speed is fixed, it is unclear that at which height wind speed is fixed. Is it U10, that is the wind speeds at 10 m above the sea surface? Please clarify these details if possible.**

*This piece of text is rewritten:(Lns. 375-382)*
It was indicated above that an initial wave field is assigned as superposition of linear modes of which amplitudes are calculated with JONSWAP spectrum with initial peak wave number $k_p^0 = 100$.The initial value $U / c_p^0 = 6$ was chosen, i.e., a ratio of the nondimensional wind speed at height of half of initial peak wave length $\lambda_0 / 2 = 2\pi / 100$ and the phase speed $c_p^0 = \left(k_p^0\right)^{-1/2}$ is equal to 6. Such a high ratio corresponds to initial stages of wave development. Wind velocity $6c_p^0$ remains constant during all time of integration.

**L218-219: \Since there are no waves in spectral models, no local criteria of wave breaking can be formulated."**
**Just a short comment here: Progress has been gradually made in spectral wave modelling over the past decade. One important outcome is that the wave breaking term in the state-of-art wave models now accounts for the threshold-behavior of dominant wave breaking, that is, waves won't break unless their steepness exceeds a threshold. The saturation spectrum B(f) = k3F(k) is used to quantify the local steepness of each wave component. See, for example,**
**Alves, J. H. G. M., and M. L. Banner, 2003: Performance of a Saturation-Based Dissipation-Rate Source Term in Modeling the Fetch-Limited Evolution of Wind Waves. Journal of Physical Oceanography, 33, 1274-1298.**
**Babanin, A. V., K. N. Tsagareli, I. R. Young, and D. J. Walker, 2010: Numerical Investigation of Spectral Evolution of Wind Waves. Part II: Dissipation Term and Evolution Tests. Journal of Physical Oceanography, 40 (4), 667-683.**

However there is considerable difference between phase-resolving and grid modeling. In spectral model the breaking in fact referred to entire field, and in direct models it is applied locally in physical space.

Added in Lns 439-443:

However, progress has been gradually made in spectral wave modeling over the past decade. One important outcome is that the wave breaking term in the state-of-art wave models now accounts for the threshold-behavior of dominant wave breaking, that is, waves won't break unless their steepness exceeds a threshold (Alves and Banner, 2003; Babanin et al. 2010).

**L227: \... many theoretical and laboratory investigations (e.g., Alberello et al., 2018)"**

**Alberello, A., A. Chabchoub, J. P. Monty, F. Nelli, J. H. Lee, J. Elsnab, and A. Toffoli, 2018: An experimental comparison of velocities underneath focussed breaking waves. Ocean Engineering, 155, 201-210.**

Reference is given. Ln. 446

**L287-288: It might be necessary to explain explicitly that B_ and B# are diffusion coefficients. And for L288: "the first versions" to "the first version"**

Done

**L329: \d_j+1 = vd_j" | It may be better to use the symbol _ instead of v here for the consistency with L86 where __j+1 = ___j is used.**

Corrected, Ln. 553

**L367-402: The author mentioned in L632 that N in RHS of Eq. (26) is very small. Is it possible to show the evolution of N in Fig. 2? I expect N is almost zero and does not vary with time as the nonlinear interaction only redistributes energy over spectral space and does not change the wave energy of the entire volume.**

Unfortunately, the total effect of nonlinear interaction can be not small, since interactions send energy to subgrid (which is absent) part of spectrum. It can happen both in spectral and direct models. In current calculation an integral term describing nonlinear interaction $\overline{\overline{N}}$ in Eq. (26) is small, but the magnitude of spectrum $N(r)$ is comparable with input $I(r)$ and dissipation $D_t(r)$ and $D_b(r)$ terms. Nonlinear interactions produces exchange by energy between modes. $\overline{\overline{N}}$ is the integral of the rate nonlinear interaction while, $N(r)$ is rate of nonlinear interactions averaged over angles.
Theoretically, $\overline{\overline{N}}$ is equal 0 but it is correct in infinitely large Fourier domain. Since our Fourier domain is restricted, $\overline{\overline{N}}$ is not zero, and dominantly negative (it could be sometimes positive because of errors of scheme). Anyway, $\overline{\overline{N}}$ is small. However, $N(r)$ is function of $r$ and it is not small, while integral of $N(r)$ over $r$ should be equal to $\overline{\overline{N}}$.

**L550-560: From Fig. 7 and Fig. 8 we know that the tail dissipation D(r) is compara-ble to or even higher than the breaking dissipation Db(r). Is this an expected behavior of this wave model?**

This effect depends significantly on parameters of numerical model. In extended domain the tail dissipation occurs in area of small energy, i.e. it can be small. Since current domain is not wide enough, the tail dissipation can be not small. It is the timeless problem of numerical mathematics.

**L632-675: It's very impressive that the shape of N(r) shown in Fig. 11 is in good agreement with Hasselmann's integral. It also might be useful to mention that Hasselmann's integral exhibits another positive lobe at high frequencies. See, for example, Hasselmann, S., K. Hasselmann, J. H. Allender, and T. P. Barnett, 1985: Computations and Parameterizations of the Nonlinear Energy Transfer in a Gravity-Wave Specturm. Part II: Parameterizations of the Nonlinear Energy Transfer for Application in Wave Models. Journal of Physical Oceanography, 15 (11), 1378-1392.**

The application of Hasselmann integral to high frequency part of spectrum is illegal, since the assumption of theory are not valid there: the modes are highly instable, and the tail consist of chaotically moving disturbances which do not satisfy the dispersive relation. So, the 'positive lobe' can exist but its nature is quite different.
The reference added in ln. 819

**L711-714: the step shape of the curve for $\omega_p$ could be possibly resulted from the discrete nature of wave models and the method utilized to calculate !p. Rogers et al. (2012, JTech) showed the similar step-shaped evolution of peak period Tp (see their Fig. 3). Rogers, E. W., A. V. Babanin, and D. W. Wang, 2012: ObservationConsistent Input and Whitecapping Dissipation in a Model for WindGenerated Surface Waves: Description and Simple Calculations. Journal of Atmospheric and Oceanic Technology, 29 (9), 1329-1346.**

Of course, the shifting of peak in numerical model cannot me monotonous. But in our case the heights of 'steps' is much bigger than 1. Hence, the new peak forms at different position and step-like behavior of downshifting has more complicated nature. Interesting that the same phenomenon is also observed in spectral model.
Comment inserted in Ln. 892

**Specific Comments:**
**L18: \thousands degrees" to \thousands of degrees"**
**L25-26: \hundreds and thousands periods" to \hundreds and thousands of periods"**
**L31: \capable to describe" to \capable of describing"**
**L32: \that of extreme wave generation" to \the generation of extreme waves"?**
**L32: \Chalikov, Babanin, 2016a" to \Chalikov and Babanin, 2016a"**
**L41: \in (Chalikov et al. 2014, Chalikov, 2016)" to \in Chalikov et al. (2014) and Chalikov (2016)"**
**L43: \A unique example" to \An unique example"**
**2**
**L44: \in (Chalikov and Babanin, 2014)" to \in Chalikov and Babanin (2014)"**
**L68: Please mention explicitly that _ or ' is velocity potential.**
**L112: delete \respectively"**
**L123: \so well" to \reasonably well"**
**L144: \This is why the function derived in (Chalikov and Rainchik, 2010)" to \This is another reason why the function derived in CR"**

L159: \on the contrary" to \on the other hand"

L162: \modes which amplitudes" to \modes of which amplitudes"

L171: \Tolman, Chalikov" to \Tolman and Chalikov, 1996"

L173: \to include" to \to be included"

L190: \This phenomenon well known" to \This well-known phenomenon"

L352: \'to33_ltration' to \to _ltration"

L429: \calculated by averaging ... 100 units of nondimensional time t" | This was already mentioned in L424-426. So maybe just simply say \The resulting wave spectra Sh(r) are presented in Fig. 3."

L611: Please clarify what is x in Fig. 9.

L707-709: the use of the symbols !p and kw are not correct here as the author intends to say !w.

L727: \the wavenumber of spectral peak kp" | Please use peak frequency !p for consistency with the caption of Fig. 13.

L734: \three-dimensional equations potential motion" to \three-dimensional equations of potential motion"

L756: \any observation data" to \any observational data"

L759: \which characteristics" to \of which characteristics"

L756-759: This argument appears too strong as _ measured from _eld experiments, such as the one proposed by Donelan et al. (2006), is also proved well-performed in operational forecasts/hindcasts.

L764-767: This sentence does not read well. It sounds like \This approach was quite accurate " due to some drawback/weakness. Please reword it, and the right bracket \)" is missing in the end of this sentence.

L768: \(Chalikov, Rainchik, 2014)" to \(Chalikov and Rainchik, 2014)"

L805: \(Ducroset et al. 216)" to \(Ducroset et al. 2016)". Besides, the year shown here and in the References list is not consistent with L738 \(Ducroset et al. 2017)".

All specific comments above have been taken into account with gratitude.

---

## Author Response (AR2)

Dear Dr Wells,

thank you for your attention to my paper. I have checked the paper and made some necessary corrections. I also hope the quality of the figures presented now is much better than before.

D. Chalikov